# LiveWeb-IE: A Benchmark For Online Web Information Extraction

**Seungbin Yang**♡◇∗    **Jihwan Kim**♡∗   **Jaemin Choi**♠   **Dongjin Kim**♡
**Soyoung Yang**♡    **ChaeHun Park**♡    **Jaegul Choo**♡
♡KAIST AI    ◇Letsur    ♠School of Computer Science and Engineering, Chung-Ang University
{sby99, jihvvan.kim, dj_kim, sy_yang, jchoo}@kaist.ac.kr
jaeminld@cau.ac.kr, qkrcogns2222@gmail.com

## Abstract

Web information extraction (WIE) is the task of automatically extracting data from web pages, offering high utility for various applications. The evaluation of WIE systems has traditionally relied on benchmarks built from HTML snapshots captured at a single point in time. However, this offline evaluation paradigm fails to account for the temporally evolving nature of the web; consequently, performance on these static benchmarks often fails to generalize to dynamic real-world scenarios. To bridge this gap, we introduce LiveWeb-IE, a new benchmark designed for evaluating WIE systems directly against live websites. Based on trusted and permission-granted websites, we curate natural language queries that require information extraction of various data categories, such as text, images, and hyperlinks. We further design these queries to represent four levels of complexity, based on the number and cardinality of attributes to be extracted, enabling a granular assessment of WIE systems. In addition, we propose Visual Grounding Scraper (VGS), a novel multi-stage agentic framework that mimics human cognitive processes by visually narrowing down web page content to extract desired information. Extensive experiments across diverse backbone models demonstrate the effectiveness and robustness of VGS. We believe that this study lays the foundation for developing practical and robust WIE systems[1].

## 1 Introduction

With the growth of data on the web, automatic information extraction from web pages has become crucial for a wide range of applications, such as large-scale information analysis and decision-making (Crescenzi et al., 2001; Thapelo et al., 2021; Li et al., 2023; Pichiyan et al., 2023). The conventional approach to web information extraction (WIE) has been dominated by wrapper-based methods, which define a set of extraction rules based on the structural patterns of web pages (Lerman et al., 2003; Reis et al., 2004; Omari et al., 2017). With the advent of large language models (LLMs), language-agent-based methods have been proposed, leveraging LLMs to extract information directly from web pages (UncleCode, 2024; Lorenzo Padoan, 2024). However, both approaches face significant limitations; wrapper-based methods are often brittle and require substantial human effort to create and maintain Dalvi et al. (2009), while using LLMs directly as extractors is impractical for large-scale scraping tasks due to the substantial costs incurred by processing each web page individually. To address this problem, hybrid approaches that leverage LLMs to generate reusable wrappers have emerged as a promising solution (Huang et al., 2024; Kaur, 2025).

Despite these advancements, it is unclear whether the success of WIE systems on existing benchmarks is a reliable indicator of their performance in real-world scenarios. While existing benchmarks rely on static HTML snapshots, real-world web pages frequently change their layouts and structures over time (Hao et al., 2011; Bronzi et al., 2013; Lockard et al., 2019). Given the strong dependence of WIE performance on the structural properties of web pages (Zheng et al., 2007; Wang

---

∗Equal contribution
[1]Our dataset is available in `https://github.com/sbY99/LiveWeb-IE`

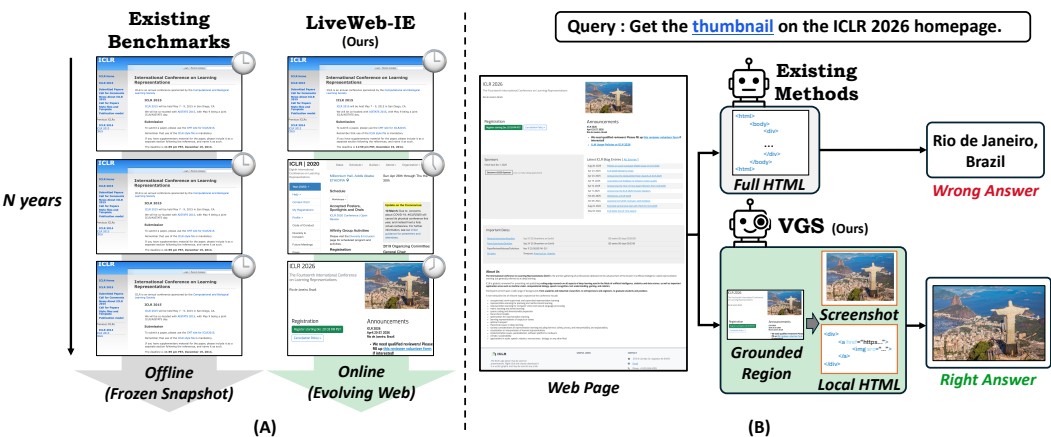

Figure 1: Overview of the conventional WIE paradigm's limitations and our solutions involving a new benchmark and an extraction method. **(A)** While existing offline benchmarks are constructed from static HTML snapshots, LIVEWEB-IE evaluates WIE systems on live websites to reflect the evolving nature of the web. **(B)** On a complex live web page, methods that process full HTML often fail, whereas VGS leverages visual cues from the rendered page for accurate information extraction.

et al., 2022), performance measured on these *offline* benchmarks, which fail to capture such temporal shifts, may not correlate with efficacy on the live websites (Mialon et al., 2023; Pan et al., 2024). Our empirical analysis in Appendix F confirms this discrepancy, showing that LLM-based wrapper generation methods suffer an average F1 degradation of over 15% when applied to the structurally evolved versions of the same websites compared to their static snapshots.

To fill this gap, we introduce LIVEWEB-IE, a benchmark for *online* web information extraction designed with four key features: **(1) Live web evaluation.** While conventional benchmarks evaluate WIE systems on static HTML snapshots, LIVEWEB-IE mandates evaluation directly on live websites. By requiring WIE systems to access the target URL, LIVEWEB-IE enables an assessment of their performance on the live web page at the time of evaluation, as shown in Figure 1-(A). **(2) Diverse and reliable websites.** LIVEWEB-IE is constructed from 15 permission-granted websites across 8 diverse domains. We select websites by considering high content stability, a crucial factor for ensuring the long-term validity of data annotation (Mialon et al., 2023). Specifically, we define this content stability as the temporal persistence of informational content, distinguishing it from layout stability. Our benchmark design ensures this content stability by curating queries for fact-based information that is unlikely to change. By targeting stable values within dynamic layouts, LIVEWEB-IE enables a robust evaluation of a system's ability to handle the web page structures at the moment of evaluation. **(3) Query-driven WIE on diverse data categories.** LIVEWEB-IE follows an on-demand IE setup, allowing users to request information through free-form natural language queries (Sekine, 2006; Jiao et al., 2023; Qi et al., 2024). In real-world scenarios, these queries are not limited to text; users often want to extract non-textual information, such as product images or event banners. However, existing benchmarks have focused on text-only extraction, failing to represent this full scope. LIVEWEB-IE addresses this gap by comprising a diverse set of manually curated queries that require WIE systems to extract various data categories, including text, images, and hyperlinks, to closely align with these real-world scenarios. **(4) Multi-dimensional task complexity.** To support a comprehensive evaluation, LIVEWEB-IE organizes extraction tasks based on two properties: the number of attributes and the cardinality of their values. This combination yields four complexity levels, from extracting a single attribute with a single value (`Type I`) to multiple attributes with lists of values (`Type IV`).

Extensive experiments on LIVEWEB-IE show that even methods with strong performance on existing benchmarks struggle significantly, especially on complex extraction tasks. We attribute this performance degradation to approaches that only rely on parsing HTML, as its inherently verbose nature makes it challenging to extract the desired information (Bevendorff et al., 2023; He et al., 2024). As the structure of web pages grows more complex over time (Vogel & Springer, 2022), the evolving nature of the web further amplifies the performance degradation of such methods.

Based on these observations, we propose Visual Grounding Scraper (VGS), a novel multi-stage framework that advances the promising paradigm of generating reusable wrappers. VGS emulates how humans find information on a web page: VGS first visually scans the web page to find the region where the information is, and then identifies the specific items within that region. This strategy bypasses the noise inherent in raw HTML by focusing on the rendered web page, thereby generating a reliable wrapper. Our method establishes state-of-the-art results not only on our challenging LIVEWEB-IE but also on existing WIE benchmarks. In summary, our contributions are as follows:

- **New Benchmark for Online WIE:** We introduce LIVEWEB-IE, a benchmark designed to overcome the limitations of *offline* evaluation by assessing WIE systems on *online* websites. It features diverse data categories, query-driven tasks, and a multi-dimensional complexity scheme.

- **Vision-Based Scraping Framework:** We propose VGS, a framework that emulates how humans find information on a web page. VGS visually identifies a relevant region on the web page and then pinpoints the specific target elements to generate accurate wrappers.

- **Extensive Evaluation:** We prove the effectiveness and robustness of VGS through extensive experiments across various backbone models, showing state-of-the-art performance on both our challenging LIVEWEB-IE and existing WIE benchmarks. Furthermore, we suggest directions for future work through human evaluations and a detailed analysis of performance.

## 2 RELATED WORK

**WIE Benchmark.** Extracting structured information from web pages is a long-standing research challenge due to the diverse and unstructured nature of web documents (Etzioni et al., 2008; Manabe & Tajima, 2015; Wang et al., 2022). This task has been evaluated using benchmarks that rely on static HTML snapshots (Hao et al., 2011; Lockard et al., 2019; San et al., 2023; Hotti et al., 2024). These datasets, while valuable, are limited by an offline evaluation paradigm, which fails to account for the evolving nature of the web. Another line of research has introduced general-purpose web benchmarks (Shi et al., 2017; Liu et al., 2018; Humphreys et al., 2022; Deng et al., 2023; Zhou et al.; He et al., 2024; Wu et al., 2025). The focus of these web agent benchmarks lies in evaluating an agent's ability to perform multi-page task completion (e.g., booking a flight) through a series of web element interactions. Consequently, these benchmarks prioritize the evaluation of sequential action execution, which is distinct from the core WIE objective that requires the precise identification and extraction of target information from within DOM structures and visual layouts. Compared with these benchmarks, as described in Table 1, LIVEWEB-IE is specifically designed to assess the task of WIE, addressing a gap between existing WIE evaluation paradigms and real-world web scraping.

**WIE Methodology.** Early WIE systems mainly studied rule-based wrapper induction systems (Kushmerick, 2000; Wu et al., 2009; Dalvi et al., 2011; Furche et al., 2016). The advent of deep learning introduced a suite of methods to better interpret HTML structures, progressing from sequence models to more advanced architectures such as FreeDOM (Lin et al., 2020) with its CNN-BLSTM text encoder and WebFormer (Guo et al., 2022), which integrates graph attention into a Transformer framework. The development of pre-trained models specialized for web data, such as MarkupLM (Li et al., 2022), and the incorporation of visual cues with models like WIERT (Li et al., 2023), further enhanced extraction accuracy by understanding HTML semantics and layouts. Recently, the advent of LLMs has introduced language-agent-based systems that leverage reasoning capabilities for direct extraction (UncleCode, 2024; Lorenzo Padoan, 2024). While flexible, these approaches are impractical for large-scale scraping tasks due to the significant latency incurred by performing inference on each page. To address this challenge, Huang et al. (2024) and Kaur (2025) proposed hybrid approaches that utilize LLMs to generate reusable wrappers. The novelty of these works lies in their HTML-based pre-processing techniques, which prune the HTML content to filter out irrelevant noise for XPath generation. However, even the state-of-the-art methods struggle on complex web pages, primarily due to an over-reliance on HTML. To fill this gap, we propose VGS, a vision-grounded framework that marks a methodological shift from prior approaches by emulating the human cognitive process and using visual information to sequentially filter out irrelevant noise.

Table 1: Comparison between LIVEWEB-IE and other web information extraction benchmarks. The query in ClosedIE corresponds to attributes, and in OpenIE to predicates. † denotes the page groups, clusters of structurally similar web pages within the website.

| Benchmarks | Task | Modality | Online | # Domain | # Website | # Query |
|---|---|---|---|---|---|---|
| SWDE (Hao et al., 2011) | Closed IE | Text | ✗ | 8 | 80 | 32 |
| WEIR (Bronzi et al., 2013) | Closed IE | Text | ✗ | 4 | 40 | 32 |
| DS1 (Omari et al., 2017) | Closed IE | Text | ✗ | 4 | 30 | 11 |
| Expanded SWDE (Lockard et al., 2019) | Open IE | Text | ✗ | 3 | 21 | 748 |
| PLAtE (San et al., 2023) | Closed IE | Text | ✗ | 1 | 43 | 3 |
| LIVEWEB-IE (Ours) | On-Demand IE | Text & Image | ✓ | 8 | 15 (46†) | 342 |

## 3 LIVEWEB-IE

In this section, we present LIVEWEB-IE, a benchmark designed to facilitate the evaluation of WIE systems that operate on live websites. Conventional benchmarks that rely on static HTML snapshots, often captured years in the past, fail to represent the current structural properties of web pages. As WIE performance is highly dependent on these properties, it is crucial to evaluate systems against the most current version of a website. LIVEWEB-IE addresses this gap by mandating that systems access the target URL during evaluation. This protocol forces the system to process the web page's DOM structure as it exists at the moment of testing. To maintain the validity of this evaluation over time, our benchmark is carefully designed to target ground-truth values that are factually stable, even as the website's layout and structure evolve. We first define the task (§ 3.1), then detail the dataset construction process (§ 3.2), and conclude with a statistical analysis of the benchmark (§ 3.3).

### 3.1 TASK DEFINITION

Formally, given a target URL $U$ and a natural language query $Q$ that requests the extraction of target attributes $\mathcal{A} = \{a_1, \ldots, a_k\}$, the objective is to extract the set of ground-truth values $\mathcal{V} = \{v_1, \ldots, v_k\}$. The WIE system must infer the set of target attributes $\hat{\mathcal{A}} = \{\hat{a}_1, \ldots \hat{a}_k\}$ from the query $Q$ before extracting their corresponding values. Each value $v_i \in \mathcal{V}$ that corresponds to an attribute $a_i \in \mathcal{A}$ can be a single item or a list of items according to the task type:

- Type I: $o = \{(a_1, v_1)\}$, where $|\mathcal{A}| = 1$ and $v_1$ is a single item.
- Type II: $o = \{(a_1, v_1), \ldots, (a_k, v_k)\}$, where $|\mathcal{A}| > 1$ and each $v_i$ is a single item.
- Type III: $o = \{(a_1, v_1)\}$, where $|\mathcal{A}| = 1$ and $v_1$ is a list of items.
- Type IV: $o = \{(a_1, v_1), \ldots, (a_k, v_k)\}$, where $|\mathcal{A}| > 1$ and each $v_i$ is a list of items.

The performance is evaluated by first aligning the inferred attributes $\hat{\mathcal{A}}$ with the ground-truth attributes $\mathcal{A}$, and then comparing the extracted value $\hat{v}_i$ with the ground-truth value $v_i$. This query-based information extraction setup reflects practical scenarios by aiming to extract user-desired information that is often not covered by conventional ontologies (Jiao et al., 2023; Qi et al., 2024).

### 3.2 DATASET CONSTRUCTION

As shown in Figure 2, we construct LIVEWEB-IE in four stages: website selection, page grouping, data annotation, and human verification. Further details are described in Appendix C.

**Website Selection.** Our website selection process follows two core principles: ensuring benchmark validity and adhering to the ethical standards for data acquisition. To ensure diversity, we select representative websites from a wide range of domains, such as academic, e-commerce, and sports. A key criterion for selection is the stability and reliability of the websites, ensuring that their content remains consistent over time. For instance, a web page recording the results of the 2022 World Cup final between Argentina and France contains factual information that is unlikely to be changed. By ensuring the content stability, we guarantee the benchmark validity and enable reproducible evaluation (Mialon et al., 2023; Guo et al., 2024; Pan et al., 2024). For ethical data sourcing, we implement a rigorous three-stage validation process. First, we review `robots.txt` files of each site to check crawling policies. Second, we screen the Terms of Use to confirm that data usage

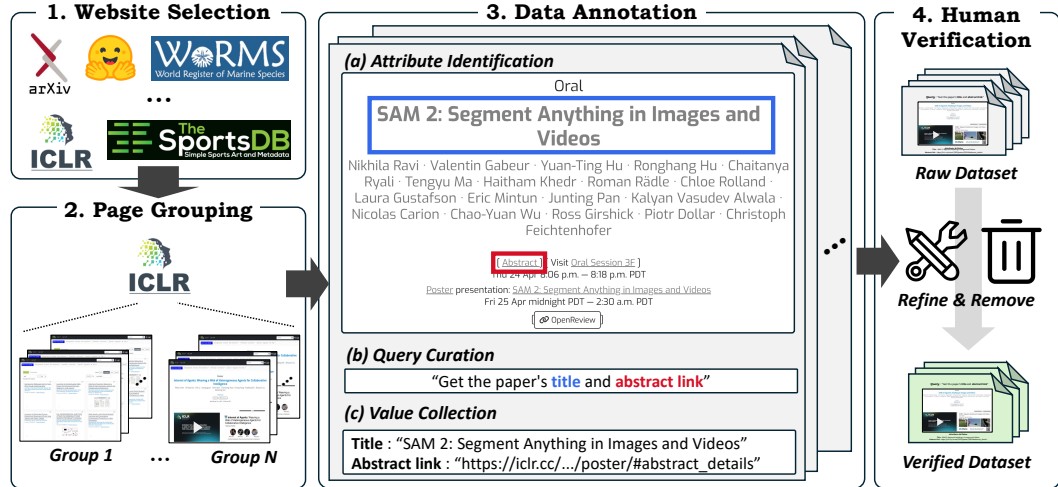

Figure 2: Dataset construction pipeline for LIVEWEB-IE. We first select a diverse set of websites and group the web pages within each website by layout. We then annotate the attributes, queries, and values for each page group, followed by a human verification process to ensure data quality.

for research is permissible. Finally, we directly contact the website administrators to obtain explicit consent. This systematic approach guarantees the ethical and legal integrity of our benchmark.

**Page Grouping.** Following the website selection process, we perform page grouping, which involves clustering web pages within each website that exhibit similar layouts. This approach distinguishes our work from conventional WIE benchmarks, which are typically constructed using web pages with uniform structural patterns. By categorizing the structural variations within each website, we enhance the structural diversity of the benchmark and facilitate an evaluation of the ability to handle the varied page layouts. A detailed description of each website is provided in Appendix H.

**Data Annotation.** For each page group, we manually curate natural language queries that request the extraction of specific attributes. Each of the five authors annotates queries for three distinct websites, following a structured procedure: First, annotators identify all potential attributes that can be extracted from a given page group, selecting only those corresponding values that are unlikely to change over time. This criterion guarantees the long-term validity of the evaluation and eliminates the need for continuous human intervention to update ground-truth values. Furthermore, we select these attributes to encompass images (e.g., fanart) and hyperlinks (e.g., profile link) alongside text, reflecting the diverse nature of real-world user queries. Each identified attribute is then categorized by whether its value is a single item or a list of items, which correspond to `Type I` and `Type III`, respectively. To construct tasks for `Type II` and `Type IV`, these attributes are combined while minimizing redundancy. Then, we curate a query in various formats to instruct the extraction of the selected attribute(s). Following the query annotation process, we collect the corresponding ground-truth values for each query. To efficiently handle the large-scale data extraction process, we develop scraping scripts tailored for each page group. These scripts automatically parse the HTML content to extract the values corresponding to the attributes. For non-textual attributes, we functionally integrate by collecting their source values, such as `@src` for images and `@href` for hyperlinks, as the ground truth.

**Human Verification.** To ensure the quality of LIVEWEB-IE, we implement a rigorous verification protocol. Each annotated sample is cross-validated by two annotators who are not involved in its creation. This review process focuses on three criteria: (1) whether the target attributes are clearly identifiable on the web page and correctly classified according to the task type, (2) whether the query is temporally stable and accurately specifies the desired information, and (3) whether the annotated ground-truth values are correct. Samples that failed to meet these standards are either refined or removed by the verifiers. The details of dataset construction are provided in Appendix C.

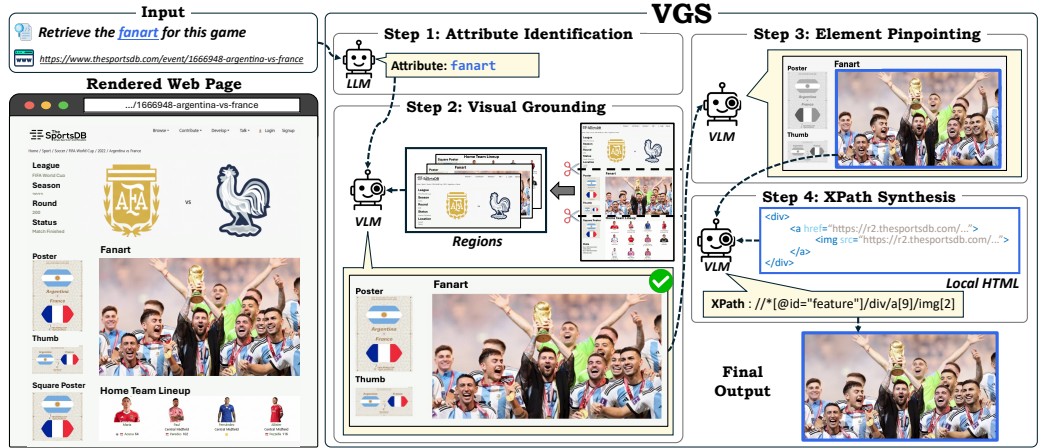

Figure 3: The framework of VGS. It sequentially narrows the observation space, from identifying target attributes, grounding the region, pinpointing the exact items, and generating the XPaths.

## 3.3 DATASET ANALYSIS

Through the multi-stage data construction process, we construct LIVEWEB-IE, a benchmark designed for evaluating WIE systems on live websites. The final dataset spans 8 domains, 15 websites, and 46 distinct page groups, comprising 342 natural language queries and 97 unique attributes. The distribution of the queries across task type and data categories is illustrated in

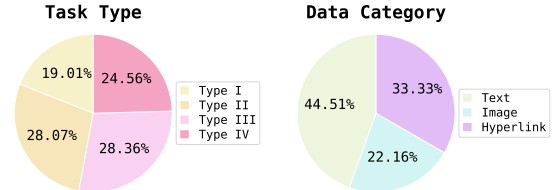

Figure 4: The task type and data category distribution.

Figure 4. Detailed statistics for each website are provided in Table 7 to Table 9.

## 4 VGS: VISUAL GROUNDING SCRAPER

In this section, we introduce Visual Grounding Scraper (VGS), a novel agentic framework designed to generate robust wrappers. As shown in Figure 3, we employ a multi-stage process that mimics the human cognitive process for seeking information on a web page. First, VGS decomposes a natural language query into a set of target attributes (§ 4.1). For each attribute, it identifies a relevant region on the web page (§ 4.2), pinpoints the exact items of interest using bounding boxes (§ 4.3), and finally synthesizes generalizable XPaths using the highly focused visual and structural context (§ 4.4). This strategy accurately identifies the information required to generate wrappers by sequentially narrowing the observation space. The overall workflow of VGS is illustrated in Figure 3.

### 4.1 ATTRIBUTE IDENTIFICATION

The initial step is to decompose the user query $Q$ into a structured set of target attributes $\hat{\mathcal{A}} = \{\hat{a}_i\}_{i=1}^k$ using an LLM guided by an attribute identification instruction $I_a$:

$$\hat{\mathcal{A}} = \text{LLM}(I_a, Q) \tag{1}$$

This step translates the unstructured query into a concrete set of extraction goals, thereby establishing a clear objective that guides the entire subsequent process.

### 4.2 VISUAL GROUNDING

Following the attribute identification step, we perform visual grounding to locate the relevant web page regions for each target attribute. The web page $P$ for a target URL $U$ is conceptualized as a sequence of vertical regions $\mathcal{R} = \{r_j\}_{j=1}^n$, where each region $r_j$ corresponds to a screenshot with

a fixed width and height. Since different attributes can be located in distinct regions of the web page, visual grounding is executed individually for each attribute $\hat{a}_i \in \hat{A}$. Given a visual grounding instruction $I_g$, a vision language model (VLM) evaluates the regions $\mathcal{R}$ to identify the pertinent region $r'_i$ for each attribute $\hat{a}_i$. Through this evaluation, a mapping $\mathcal{R}'$ is established, linking each predicted attribute $\hat{a}_i$ with its corresponding visually grounded region $r'_i$:

$$\mathcal{R}' = \{(\hat{a}_i, r'_i)\}_{i=1}^k, \text{ where } r'_i = \text{VLM}(I_g, \mathcal{R}, \hat{a}_i) \tag{2}$$

This step prunes the observation space by isolating the most relevant regions of the web page. Consequently, subsequent stages can operate on a condensed and contextually rich subset of the page.

## 4.3 ELEMENT PINPOINTING

Once the region containing the value for a target attribute is identified, we pinpoint the exact location of the value to identify the items required for XPath generation. For a precise identification of attribute-relevant items, we adopt a two-stage approach. First, we generate a set of candidate bounding boxes $\mathcal{B}_i = \{b_{i,j}\}_{j=1}^m$, around potential items of interest within $r'_i$. The strategy for generating these candidates is adapted based on the type of the attribute $\hat{a}_i$. For attributes whose values are directly rendered as text, we visually scan the region $r'_i$ and identify all relevant text items using the VLM. For non-textual attributes such as images or hyperlinks, we identify candidate items based on tags relevant to the attribute type (e.g., `` tag for images). To mark these candidates in the region, we employ Set-of-Mark Prompting (Yang et al., 2023). Specifically, we use a JavaScript tool to generate and overlay bounding boxes with unique numerical labels onto the candidate items within the region. The overlay function $\mathcal{O}$ applies the candidate bounding boxes $\mathcal{B}_i$ to the region $r'_i$, generating the visually marked region $r^*_i = \mathcal{O}(\mathcal{B}_i, r'_i)$. Then, the VLM processes the marked region $r^*_i$ based on the pinpointing instruction $I_p$ to select the subset of bounding boxes $\mathcal{B}^*_i \subseteq \mathcal{B}_i$ corresponding to the actual values of attribute $\hat{a}_i$. This pinpointing process is formally defined as:

$$\mathcal{B}^* = \{\mathcal{B}^*_i\}_{i=1}^k, \text{ where } \mathcal{B}^*_i = \text{VLM}(I_p, r^*_i, \hat{a}_i) \tag{3}$$

We finalize the localization process by pinpointing the values corresponding to the target attributes. The resulting set of bounding boxes supplies the evidence for the subsequent XPath synthesis.

## 4.4 XPATH SYNTHESIS

In the final stage, we synthesize reusable XPaths using the set of validated bounding boxes $\mathcal{B}^*$. For each bounding box $b^*_{ij} \in \mathcal{B}^*_i$, we first identify the primary Document Object Model (DOM) element $e_b$ corresponding to the coordinates of the box. To provide sufficient context while minimizing noise, it then extracts a local HTML segment $h_b$, consisting of $e_b$ and neighboring elements within a distance $d$. The VLM synthesizes a generalizable XPath $x_i$ for an attribute $\hat{a}_i$ using an XPath synthesis instruction $I_x$, the set of all relevant local HTML segments $\mathcal{H}_i = \{h_{b^*_{ij}}\}_{j=1}^l$, the target attribute $\hat{a}_i$ and the further pinpointed region $\hat{r}_i = \mathcal{O}(\mathcal{B}^*_i, r'_i)$. The set of XPaths $\mathcal{X}$ that constitutes the final wrapper is formally defined as:

$$\mathcal{X} = \{x_i\}_{i=1}^k, \text{ where } x_i = \text{VLM}(I_x, \mathcal{H}_i, \hat{r}_i, \hat{a}_i) \tag{4}$$

By grounding the generation in both visual evidence and localized information, the resulting XPath achieves high precision and generalizability. Each XPath $x_i$ within the set of XPaths $\mathcal{X}$ is used to generate the predicted value $\hat{v}_i$ for its corresponding attribute $\hat{a}_i$.

# 5 EXPERIMENTS

## 5.1 EXPERIMENTAL SETTING

**Baselines.** We compare VGS against methods that leverage LLMs to generate reusable wrappers. We adopt Chain-of-Thought (CoT) (Wei et al., 2022), Reflexion (Shinn et al., 2023), and Auto-Scraper (Huang et al., 2024) as baselines. CoT generates an XPath in a single pass, whereas Reflexion operates iteratively to refine its output based on execution failures. AutoScraper also follows an iterative process, but proactively simplifies the web page by pruning the DOM tree. We also report the performance of a human evaluation conducted by six experts with relevant backgrounds. Details for human evaluation are available in Appendix B.3.

Table 2: Main experimental results on LIVEWEB-IE, measured by Precision (P), Recall (R), and F1 score (F1). The best score for each metric is highlighted in **bold**.

| Models | Method | Type I | | | Type II | | | Type III | | | Type IV | | | Overall | | |
|---|---|---|---|---|---|---|---|---|---|---|---|---|---|---|---|---|
| | | P | R | F1 | P | R | F1 | P | R | F1 | P | R | F1 | P | R | F1 |
| *Proprietary Models* | | | | | | | | | | | | | | | | |
| Gemini-2.5-Flash | COT | 32.04 | 49.01 | 34.36 | 32.19 | 54.80 | 35.17 | 9.66 | 13.99 | 8.19 | 10.16 | 10.90 | 7.40 | 20.36 | 31.33 | 20.53 |
| | Reflexion | 33.00 | 48.31 | 35.96 | 39.87 | 39.59 | 39.40 | 10.29 | 13.00 | 8.30 | 12.83 | 23.11 | 8.67 | 23.53 | 29.66 | 22.39 |
| | AutoScraper | 36.52 | 42.35 | 37.25 | 31.47 | **55.76** | 34.39 | 10.94 | 18.61 | 10.23 | 14.36 | 23.75 | 9.66 | 22.41 | 34.82 | 22.02 |
| | VGS | 47.83 | 55.85 | 49.02 | 42.36 | 53.76 | 44.82 | 43.61 | 43.49 | 42.92 | 38.60 | 38.32 | 38.13 | 42.81 | 47.46 | 43.44 |
| GPT-4o-mini | COT | 32.15 | 41.08 | 33.41 | 31.86 | 54.23 | **34.80** | 8.88 | 18.45 | 8.76 | 12.22 | 22.01 | 8.26 | 20.58 | 33.66 | 20.63 |
| | Reflexion | 31.97 | 38.15 | 33.17 | 24.84 | 31.37 | 25.95 | 8.26 | 9.09 | 7.60 | 9.68 | 10.38 | 7.05 | 17.75 | 21.18 | 17.47 |
| | AutoScraper | 35.10 | 43.23 | 36.25 | 31.14 | **55.17** | 34.03 | 11.74 | 19.26 | 11.05 | 13.67 | 22.62 | 9.20 | 22.10 | 34.73 | 21.85 |
| | VGS | 49.11 | 57.55 | 50.36 | 32.37 | 46.30 | 33.32 | 33.87 | 34.58 | 33.95 | 30.29 | 30.72 | 29.68 | 35.48 | 41.28 | 35.85 |
| GPT-4o | COT | 45.96 | 52.83 | 47.54 | 39.53 | 49.51 | 40.84 | 12.14 | 11.44 | 8.15 | 11.20 | 16.70 | 7.24 | 26.03 | 31.27 | 24.60 |
| | Reflexion | 47.75 | 50.75 | 48.75 | 39.46 | 43.54 | 39.64 | 15.99 | 10.34 | 10.24 | 7.24 | 3.69 | 3.76 | 26.47 | 25.71 | 24.22 |
| | AutoScraper | 53.23 | 59.52 | 55.22 | 41.93 | 52.79 | 42.65 | 13.95 | 12.10 | 9.10 | 12.60 | 13.37 | 6.92 | 28.94 | 32.86 | 26.76 |
| | VGS | **64.51** | **69.78** | **65.87** | **44.67** | 55.49 | **46.35** | **45.25** | **45.73** | **45.38** | **41.33** | **41.86** | **41.50** | **47.78** | **52.09** | **48.58** |
| *Open-Source Models* | | | | | | | | | | | | | | | | |
| Gemma-3-4B | COT | 12.14 | **25.23** | 14.62 | 3.81 | 10.11 | 4.71 | 5.63 | 7.22 | 5.90 | 0.60 | 1.19 | 0.79 | 5.11 | 9.98 | 5.98 |
| | Reflexion | 12.19 | 15.94 | 12.45 | 10.54 | 30.00 | 13.61 | 9.57 | 32.23 | 10.99 | 4.19 | 4.79 | 4.21 | 9.01 | 21.78 | 10.35 |
| | AutoScraper | 14.30 | 18.06 | 14.95 | 11.54 | **30.19** | 14.04 | 9.67 | **35.85** | 10.27 | **7.08** | 17.94 | **7.61** | 10.44 | **26.48** | 11.57 |
| | VGS | 19.97 | 21.47 | 20.10 | 21.03 | 27.66 | 22.33 | 15.02 | 24.92 | 15.70 | 4.98 | 29.53 | 7.19 | 15.18 | 26.17 | 16.30 |
| Gemma-3-27B | COT | 31.53 | 47.08 | 32.63 | 17.06 | 46.96 | 20.07 | 9.56 | 17.39 | 10.69 | 6.15 | 17.24 | 7.29 | 15.01 | 31.29 | 16.65 |
| | Reflexion | 36.71 | 44.31 | 37.84 | 15.24 | 25.73 | 16.35 | 10.16 | 17.95 | 12.03 | 9.28 | 15.83 | 9.27 | 16.42 | 24.63 | 17.47 |
| | AutoScraper | 33.50 | 45.85 | 34.63 | 19.51 | 53.35 | 22.87 | 9.81 | 16.58 | 10.27 | 10.92 | 13.36 | 10.31 | 17.30 | 31.67 | 19.04 |
| | VGS | 36.99 | 50.00 | 38.72 | 23.75 | 61.27 | 28.46 | 27.90 | 31.82 | 29.73 | 33.58 | 29.65 | 28.46 | 29.87 | 43.02 | 30.79 |
| Qwen-2.5-7B | COT | 34.41 | 43.21 | 35.08 | 9.44 | 16.25 | 9.82 | 4.37 | 15.85 | 4.98 | 2.28 | 8.16 | 3.45 | 10.99 | 19.27 | 11.67 |
| | Reflexion | 36.56 | 46.98 | **38.12** | 12.19 | 19.75 | 13.30 | 5.64 | **23.71** | 7.76 | 5.03 | 16.55 | 5.50 | 13.22 | 25.25 | 14.53 |
| | AutoScraper | 36.00 | 42.08 | 36.75 | 19.41 | 41.57 | 19.53 | 7.35 | 11.64 | 7.50 | 3.77 | **21.55** | 5.90 | 15.30 | 28.25 | 16.04 |
| | VGS | 37.99 | 47.36 | 37.31 | 23.00 | 43.67 | 23.50 | 18.39 | 16.63 | 16.64 | 13.07 | 15.28 | 13.58 | 21.60 | 30.23 | 21.74 |
| Qwen-2.5-32B | COT | 28.83 | 42.59 | 30.78 | 32.49 | 37.36 | 32.08 | 7.20 | 16.11 | 6.23 | 4.04 | 10.67 | 4.61 | 17.63 | 25.76 | 17.74 |
| | Reflexion | 37.30 | 44.91 | 37.57 | 30.90 | **48.48** | 33.05 | 8.47 | 15.02 | 6.72 | 7.53 | 13.38 | 7.98 | 20.01 | 29.68 | 20.28 |
| | AutoScraper | 40.63 | 44.72 | 41.07 | 35.42 | 46.48 | 36.20 | 7.11 | 14.63 | 8.37 | 5.48 | 9.27 | 5.22 | 21.02 | 27.97 | 21.61 |
| | VGS | 46.85 | 52.45 | 46.76 | 39.78 | 40.07 | 37.83 | 30.33 | 30.61 | 29.90 | 28.09 | 38.31 | 28.67 | 35.57 | 39.31 | 35.05 |
| Qwen-2.5-72B | COT | 35.77 | 57.93 | 39.98 | 30.18 | 45.69 | 33.87 | 9.48 | 19.62 | 7.53 | 12.42 | 18.75 | 11.49 | 21.01 | 34.02 | 22.06 |
| | Reflexion | 30.32 | 33.69 | 31.33 | 30.15 | 35.29 | 31.38 | 10.69 | 8.96 | 8.42 | 6.53 | 7.94 | 4.95 | 18.86 | 20.79 | 18.36 |
| | AutoScraper | 38.03 | **61.04** | 41.71 | 33.53 | 51.69 | 37.61 | 10.05 | 18.33 | 8.62 | 9.83 | 17.69 | 9.59 | 21.90 | 35.64 | 23.28 |
| | VGS | 48.09 | 53.96 | 48.63 | 37.38 | 60.63 | 41.10 | 35.30 | 39.43 | 36.53 | 30.42 | 31.91 | 31.06 | 37.12 | 46.30 | 38.77 |
| *Human* | | 83.31 | 92.40 | 87.13 | 85.56 | 94.05 | 86.06 | 89.30 | 89.50 | 89.04 | 82.74 | 87.51 | 84.14 | 85.24 | 90.86 | 86.60 |

**Models.** We employ a diverse set of both proprietary and open-source models. For proprietary models, we use Gemini-2.5-Flash (Comanici et al., 2025), GPT-4o-mini, and GPT-4o (Hurst et al., 2024). For open-source models, we utilize Gemma-3-4B/27B-Instruct (Team et al., 2025) and Qwen-2.5-7B/32B/72B-Instruct (Yang et al., 2024a). Since the Qwen-2.5-Instruct models are text-only, we employ their corresponding vision-language counterparts of the same size, Qwen-2.5-VL-7B/32B/72B-Instruct (Bai et al., 2025), for the visual analysis steps within our method. As the other models are inherently multi-modal, we use them consistently for all stages of our method.

**Evaluation.** We evaluate how effectively a WIE system identifies the target attributes from a natural language query and extracts their corresponding values. First, we determine if the WIE system correctly identifies the target attributes. To account for the semantic flexibility of natural language, we employ an LLM-based alignment strategy, using GPT-4o to match the inferred attributes with their ground-truth counterparts (Jiang et al., 2024; Chen et al., 2024). Inferred attributes that do not align are considered an extraction failure. Subsequently, for each aligned attribute, we compare the extracted values against the ground-truth values. In line with existing evaluation schemes for WIE, we measure performance using precision, recall, and F1 score (Hao et al., 2011; Lockard et al., 2019; Huang et al., 2024). Further details on the evaluation setup are available in Appendix B.

**Implementation Details.** All experiments are conducted in a zero-shot setting due to the context length limitations of the models. As the baseline methods do not account for interaction with the live web page, we provide the HTML of a given web page as input. For each sample, a set of XPaths is generated from the first web page in a group and then applied to all other pages within that group

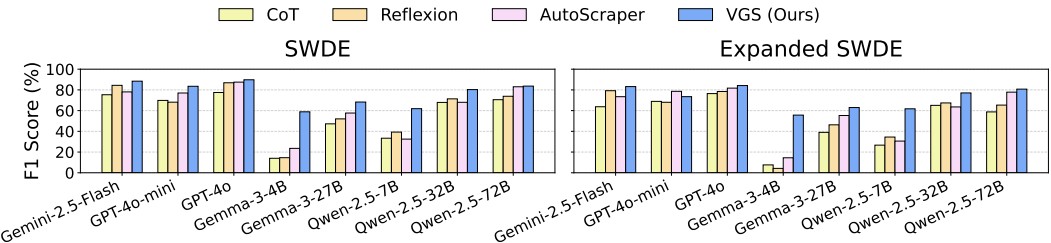

Figure 5: F1 score comparison on existing web information extraction benchmarks. VGS generally outperforms baselines across different backbone models. Full results are shown in Table 5

to extract the target values. We use Playwright [2] for rendering and navigating web pages, as well as for executing the generated XPaths. Further details on experiments are available in Appendix A.

## 5.2 MAIN RESULTS

Table 2 presents the main results for VGS and baseline methods across various backbone models, showing that VGS generally outperforms the baselines across all task types. The performance gap between VGS and the baselines is particularly pronounced on the `Type III` and `Type IV` tasks, which require the extraction of multiple attributes. Specifically, when using GPT-4o as the backbone model, VGS outperforms AutoScraper by a significant margin of 36.28% and 34.58% in F1 score on `Type III` and `Type IV` tasks, respectively. This indicates that our vision-grounded approach to identifying the precise target elements facilitates the generation of accurate XPaths on live websites. It is noteworthy that even when using the powerful backbone models, GPT-4o and Qwen-2.5-72B, the best-performing VGS falls short of human performance by 38.02% and 47.83% in overall F1 score, respectively. This finding demonstrates that our dataset is a challenging benchmark and underscores the need for further advancements to handle live web environments.

## 5.3 ABLATION STUDY

VGS filters out irrelevant information by progressively narrowing down the observation space. This process is driven by two sequential steps: visual grounding and element pinpointing. To analyze the contribution of each step, we conduct an ablation study. As shown in Table 3, the removal of any component results in a performance drop. Notably, when using Qwen-2.5-72B as the backbone, removing both components simultaneously causes a performance

Table 3: Ablation study on visual grounding and element pinpointing. We report the F1 score.

| Grounding | Pinpointing | GPT-4o | Qwen-2.5-72B |
|:---:|:---:|:---:|:---:|
| ✓ | ✓ | **48.58** | **38.77** |
| ✓ | ✗ | 43.89 (↓ 4.69%) | 34.72 (↓ 4.05%) |
| ✗ | ✓ | 45.62 (↓ 2.96%) | 37.06 (↓ 1.71%) |
| ✗ | ✗ | 43.17 (↓ 5.41%) | 31.18 (↓ 7.59%) |

drop of 7.59%, a more pronounced degradation than removing either component individually. These findings validate the effectiveness of our sequential, vision-based filtering approach.

## 6 DISCUSSION

### 6.1 EXPERIMENTS OVER EXISTING BENCHMARK

To demonstrate the robustness of our proposed method, we evaluate on existing WIE benchmarks, SWDE (Hao et al., 2011) and Expanded SWDE (Lockard et al., 2019). Following the evaluation protocol of Huang et al. (2024), we prompt each method to extract the target attributes using natural language instructions. These benchmarks only focus on preserving textual content and the basic DOM structure, often resulting in a rendered web page with visual artifacts, such as overlapping text and expired images (Yang et al., 2024b). Since our methodology operates on screenshots of rendered web pages, we manually exclude samples exhibiting significant rendering issues. Further details are provided in Appendix B.2. Our experimental results demonstrate that VGS generally achieves

---

[2]https://playwright.dev/

strong performance on both benchmarks, as shown in Figure 5. Notably, VGS exhibits a significant performance improvement over the baselines when utilizing small-sized backbone models, such as Gemma-3-4B and Qwen-2.5-7B. The results confirm the robustness and effectiveness of our vision-grounded approach, validating its effectiveness even on traditional benchmarks.

## 6.2 PERFORMANCE ANALYSIS BY DATA CATEGORY

LIVEWEB-IE incorporates non-textual data, such as images and hyperlinks, to reflect the diverse requirements of practical WIE scenarios. We analyze how performance varies across different data categories. Figure 6 illustrates the performance when using GPT-4o and Qwen-2.5-72B as backbone models. A key observation is that both baseline methods and VGS exhibit a performance degradation when extracting non-textual data. While

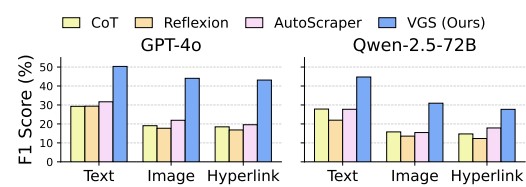

Figure 6: F1 score comparison of VGS and baseline methods across three data categories.

VGS obtains relatively high F1 scores of 44.05% for images and 43.13% for hyperlinks using the GPT-4o, these results indicate that there is still room for improvement. The difficulty of non-textual information extraction suggests that a direction for future research is to improve the capability to extract information that is not explicitly represented.

## 6.3 ERROR ANALYSIS

To better understand the failure cases of VGS, we manually analyze 100 random samples for each of the Qwen-2.5-7B/32B/72B, GPT-4o-mini, and GPT-4o models. The errors are categorized into four main failure types, corresponding to the main steps of VGS: misinterpreting the target attribute, failing to ground the region, incorrectly pinpointing the target element, and generating wrong XPaths. Figure 7 presents the distribution of these error types. Our analysis reveals that a significant portion of errors occurs during the stages that leverage visual modality to identify information corresponding to the

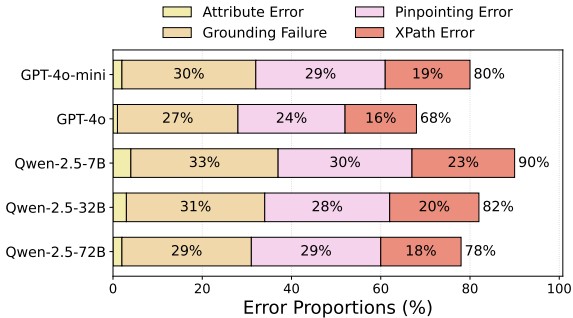

Figure 7: Distribution of error types for VGS across different backbone models.

target attributes. While leveraging visual context to filter out irrelevant information proves beneficial for generating robust XPaths, precisely identifying the target information remains a key challenge.

## 7 CONCLUSION

In this work, we introduce LIVEWEB-IE, a benchmark designed to evaluate the capabilities of web information extraction systems under the online web environment. LIVEWEB-IE features a collection of live websites and natural language queries designed to cover four levels of complexity. Our benchmark reveals that even methods exhibiting strong performance on existing benchmarks struggle to generalize to live websites. To address this, we propose VGS, a multi-stage framework that leverages visual context to accurately identify the elements required to generate XPath by progressively filtering out irrelevant information. Experiments show that VGS outperforms strong baselines on both LIVEWEB-IE and existing benchmarks. Our work sets a new paradigm for evaluation and methodology, driving progress toward more practical web information extraction systems.

ACKNOWLEDGEMENTS

This work was supported by the Starting growth Technological R&D Program(TIPS Program, (No. RS-2023-00272605)) funded by the Ministry of SMEs and Startups(MSS, Korea) in 2023; Institute for Information & communications Technology Planning & Evaluation(IITP) grant funded by the Korea government(MSIT) (RS-2019-II190075, Artificial Intelligence Graduate School Program(KAIST)); and the National Research Foundation of Korea(NRF) grant funded by the Korea government(MSIT) (No. RS-2025-00555621).

ETHICS STATEMENT

Ethical considerations are essential to this study, particularly concerning the construction of the LIVEWEB-IE benchmark. Our protocol for website selection is guided by the principle of respecting website operators and user privacy. For each of the 15 websites that constitute our benchmark, we undertake a multi-step approval process: (1) we first review their `robots.txt` files and Terms of Use to ensure our research activities are not in violation of their stated policies. (2) Following this, we proactively contact the administrators of each website to explain the purpose of our research and formally request permission to include their website in our benchmark. We proceed only after receiving explicit consent. The natural language queries in LIVEWEB-IE are intentionally curated to extract publicly available information. Our data collection and annotation processes are designed to minimize any risk of exposing private data. We will release our benchmark under a license that restricts its application to research purposes only, accompanied by clear guidelines.

REPRODUCIBILITY STATEMENT

To ensure reproducibility, we provide our LIVEWEB-IE and all experimental details. The dataset is accessible through an anonymized GitHub repository (`https://github.com/sbY99/LiveWeb-IE`). Appendix A documents the experimental environment and hyperparameters. Appendix B describes the evaluation protocols for our dataset and existing benchmarks. All prompts used in the experiments are provided in Appendix J.

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

# A    IMPLEMENTATION DETAILS

**Environment Setup.**    We perform experiments on a Linux server with two NVIDIA H200 GPUs. Our implementation relies on Python 3.12.11, CUDA 12.9, and the Playwright library (version 1.54.0) for web browser automation. We set a web page viewport size of $1280 \times 1100$ pixels and a neighbor distance $d$ of 2. For open-source model inference, we utilize the vLLM library (version 0.10.1.1).

**Hyperparameters.**    To ensure consistent and reproducible results across all experiments, we apply uniform generation hyperparameters to all models. We use a temperature of 0.0 and a max sequence length of 8192 for the generation hyperparameters.

**LLM-based Wrapper Generation.**    Raw HTML content contains numerous irrelevant elements that interfere with LLMs, while the full content is often too large to fit within the context length limitations of LLMs. Following Huang et al. (2024), we provide simplified HTML to LLM-based wrapper baselines. Specifically, we first remove elements that contain `<script>` and `<style>` from the DOM tree. Then, we preserve the `@class` attribute alongside essential functional attributes, including `@href` for hyperlinks and `@src` and `@alt` for images.

# B    EVALUATION DETAILS

## B.1    LIVEWEB-IE

We employ the powerful LLM (GPT-4o) to judge whether predicted attributes from WIE systems correspond to ground-truth attributes. This approach is crucial for accommodating the semantic flexibility inherent in natural language. For instance, if the ground-truth attribute is "*author profile link*" and the system predicts "*author info page link*", our evaluation method correctly identifies these as synonymous. To ensure the validity of this automated evaluation, we manually review the 50 samples where the predicted attributes were not exact literal matches to the ground truth. The judgments from GPT-4o achieve a 98% agreement rate with our manual assessments, confirming its reliability.

## B.2    EXISTING BENCHMARKS

Following Huang et al. (2024), we sample 100 web pages per website and exclude those with rendering issues such as text overlap. For the SWDE benchmark, we remove two websites, CollegeToolkit and FanHouse, from the university and NBAPlayer domains, respectively. As each website has four associated cases, this reduces the initial 320 cases by 8, resulting in a final set of 312 cases. For the Expanded SWDE benchmark, we first subsample 294 attributes by aligning relations with an established attribute set and discarding outlier cases (Huang et al., 2024). We then filter out one website with rendering problems, which removes its 14 associated attributes. This yields a final set of 280 attributes for evaluation. Figure 11 shows an example of an excluded web page.

## B.3    HUMAN EVALUATION

To establish human performance for LIVEWEB-IE, we recruit six evaluators with Bachelor's degrees in Computer Science and experience with web scraping. We select a balanced subset of 120 samples across websites and task types, distributing them equally among the evaluators. Evaluators write executable web scraping scripts according to these guidelines: (1) create executable web scraping scripts for each query on its corresponding page group, (2) for non-textual content such as images and hyperlinks, extract the value of the corresponding source attribute (e.g., `a` tag for hyperlink), and (3) avoid receiving direct help from AI assistants for generating web scraping scripts. We then execute the web scraping scripts submitted by each evaluator. The results are compared against the ground truth using the same evaluation methodology.

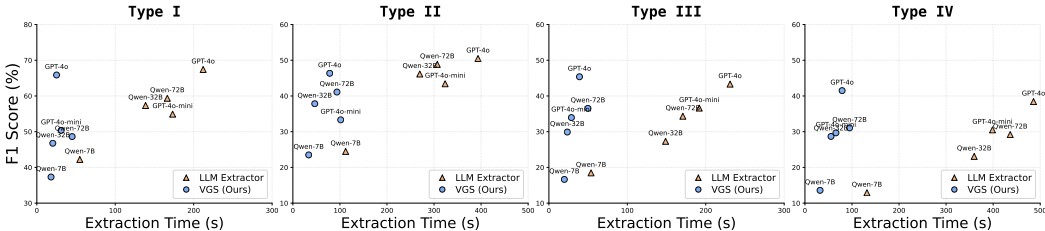

Figure 8: F1 score and extraction time comparison between VGS and a direct extraction approach across the four task types. The extraction time is measured in seconds (s) per page group.

## C    DETAILS FOR LIVEWEB-IE CONSTRUCTION

The dataset construction process involves five authors with Bachelor's degrees or higher in Computer Science. We annotate the data through attribute identification, query formulation, ground-truth value extraction, and human validation. To identify the attributes from the web page, we use the original text from the source web pages as attributes (Lockard et al., 2019). If multiple values share identical text within a page group (e.g., "*link*" referring to both league link and team link), we add descriptive prefixes to create distinct attributes such as "*league link*" and "*team link*". For web pages where attributes are not explicitly identified, we assign semantically clear attributes derived from the meaning of their associated values. When formulating queries, we create instructions that accurately specify the desired information. We design temporally stable queries that avoid time-dependent references. For example, "Extract logo information for all seasons" updates as new seasons are added. For ground-truth value extraction, we extract values according to their data categories. We extract `@src` values for images and `@href` values for hyperlinks. These correspond to target destinations and source paths as specified in HTML standards[3][4].

## D    IMPACT OF CONTEXT DISTANCE

In our XPath Synthesis stage (§ 4.4), the distance parameter $d$ defines the size of the local HTML segment used as context. This parameter manages the trade-off between providing sufficient structural information and introducing noise. To investigate its effect on WIE performance, we analyze by varying $d \in \{0, 1, 2, 3, 4\}$. As shown in Figure 9, performance degrades at the extremes, specifically at $d = 0$ and $d = 4$. We hypothesize that this is because an insufficient context ($d = 0$) leads to overly specific XPaths that lack generalizability, while an excessive context ($d = 4$) confuses the VLM with irrelevant information, leading to imprecise XPath generation.

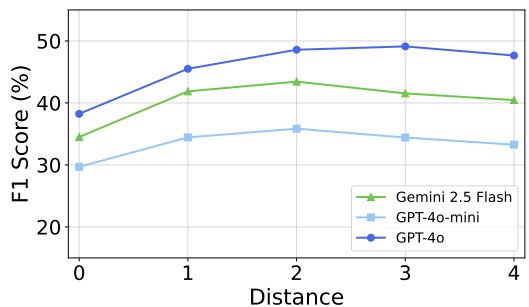

Figure 9: Performance variation across distances.

## E    COMPARISON WITH DIRECT EXTRACTION VIA LANGUAGE AGENT

Another approach for WIE involves utilizing language agents to reason directly over HTML content to extract desired information. To compare VGS with this approach, we establish a baseline that prompts an LLM to extract the target attributes directly. Figure 8 illustrates the comparison result. While LLM extractor offers high flexibility, its page-by-page reasoning process incurs substantial inference time. Moreover, the direct extraction approach is vulnerable to model context length limitations because it generates all target values directly. This leads to a significant performance drop in scenarios that require extracting a large number of values, such as `Type IV`. In contrast,

---

[3]https://html.spec.whatwg.org/multipage/text-level-semantics.html
[4]https://html.spec.whatwg.org/multipage/embedded-content.html

Table 4: Performance comparison between the SWDE-sub and SWDE-2025. The SWDE-sub refers to a snapshot manually sampled from the original SWDE dataset, where the attributes and values remain identical to the live web pages. SWDE-2025 denotes the corresponding web pages captured in November 2025. ΔF1 represents the difference in F1 scores, obtained by subtracting SWDE-sub from SWDE-2025, which highlights the consistent performance decline across all approaches.

| Model | Method | SWDE-sub | | | SWDE-2025 | | | Δ F1 |
|---|---|---|---|---|---|---|---|---|
| | | P | R | F1 | P | R | F1 | |
| GPT-4o | CoT | 95.05 | 85.0 | 82.62 | 97.94 | 70.32 | 68.95 | -13.67 |
| | Reflexion | 91.98 | 95.53 | 90.70 | 98.47 | 73.24 | 73.86 | -16.93 |
| | AutoScraper | **99.47** | 90.79 | 92.66 | **99.06** | 77.48 | 77.83 | -14.83 |
| | VGS (ours) | 98.12 | **96.27** | **94.53** | 98.21 | **78.94** | **86.38** | -8.15 |
| Qwen-2.5-72B | CoT | **99.47** | 71.58 | 73.32 | **95.08** | 57.38 | 56.33 | -16.99 |
| | Reflexion | 94.62 | 79.74 | 76.67 | 94.12 | 60.89 | 58.79 | -17.88 |
| | AutoScraper | 95.57 | 82.11 | 80.0 | 92.40 | 66.16 | 62.58 | -17.51 |
| | VGS (ours) | 93.53 | **85.59** | **83.48** | 94.05 | **76.12** | **74.85** | -8.63 |

VGS operates as a hybrid method that employs a VLM to generate reusable XPaths, offering an efficient solution for large-scale data extraction.

## F  PERFORMANCE GAP BETWEEN STATIC SNAPSHOTS AND LIVE WEB PAGES

To investigate whether the WIE performance reported on offline benchmarks generalizes to current web environments, we conduct a comparative evaluation of WIE systems on both static snapshots and their corresponding live web pages. We utilize the SWDE dataset, which consists of HTML snapshots from the early 2010s. From this dataset, we manually curate a new test set by comparing these snapshots to their corresponding current versions (November 2025). To ensure a valid comparison, we select only web pages where the target value remains present and identical to the original SWDE ground truth. This process yields a curated dataset covering 10 websites with 10 pages each, as detailed in Table 6. We then evaluate both the LLM-based wrapper generation baselines and VGS on the original snapshots and their live counterparts.

As shown in Table 4, the performance on offline snapshots fails to generalize to the current version of the same websites. With the GPT-4o model, the F1 scores for HTML-based baselines drop by an average of 15.14%, and VGS drops by 8.15%. Similarly, on the Qwen-2.5-72B model, the baselines show an average F1 degradation of 17.46%, whereas VGS degrades by 8.63%. This validates our core motivation and demonstrates the need for a benchmark akin to LIVEWEB-IE, which evaluates systems directly on live web pages. Furthermore, it suggests our vision-grounded approach is more robust to structural evolution.

## G  EFFICIENCY COMPARISON WITH LLM-BASED WRAPPER GENERATION BASELINES

To compare the efficiency of VGS against LLM-based wrapper generation baselines, we conduct an experiment across varying levels of web page complexity. We stratify the LIVEWEB-IE samples based on average HTML length, serving as a proxy for complexity. Within each complexity group, we measure the average F1 score and the mean inference time for VGS and the baselines.

As illustrated in Figure 10, VGS exhibits a higher initial latency on simpler web pages compared to the baselines. However, as structural complexity increases, the inference latency of iterative HTML-based methods (Reflexion and AutoScraper) increases substantially, eventually converging with that of VGS on highly complex pages. Crucially, VGS consistently maintains superior extraction performance across all complexity levels. Considering that robust extraction from complex, content-heavy websites represents the primary bottleneck in real-world WIE, we argue that VGS offers a practical and favorable solution for real-world deployment.

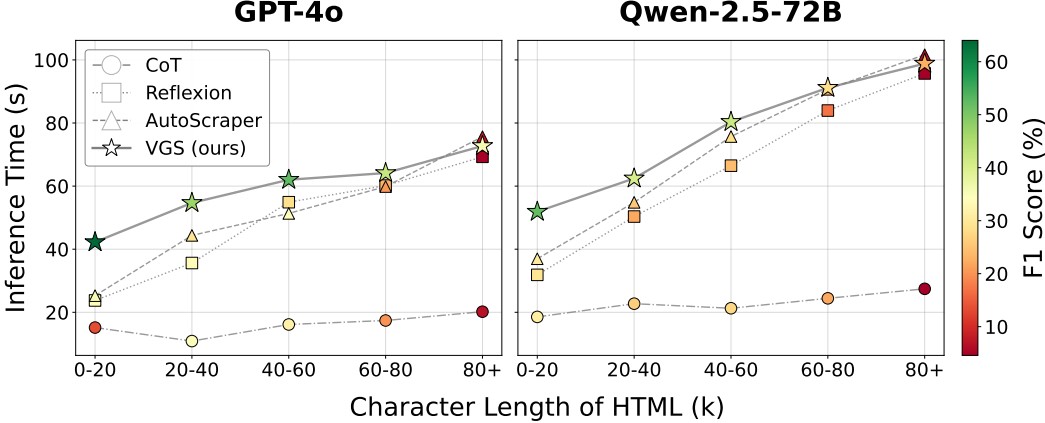

Figure 10: Efficiency comparison between VGS and LLM-based wrapper generation baselines. The character length represents the average HTML character length of the web pages for each data sample.

## H WEBSITE DESCRIPTIONS FOR LIVEWEB-IE

This section provides detailed descriptions of the websites included in LIVEWEB-IE.

### H.1 ACADEMIC

**Arxiv.** ArXiv[5] repository is a major preprint database for scientific literature, widely used in computer science. The platform hosts a vast collection of scientific articles, each accompanied by rich metadata including titles, authors, and abstracts. For this website, we constructed queries related to extracting paper titles, author names, and direct PDF links.

**Machine Learning Conferences.** The ICLR[6], ICML[7], and NeurIPS[8] platforms represent top-tier machine learning conferences with standardized academic content structures. These websites host the proceedings for each conference, including lists of accepted papers, author information, and abstracts. Example tasks include finding the abstract page for a given paper or extracting the URLs for all papers in a specific session.

**Hugging Face Daily Papers.** Hugging Face Daily Papers[9] is a community platform that aggregates and ranks recent AI research papers. The platform features a daily feed of new papers, each with community-driven features like discussion threads, like counts, and links to associated code or models. Our queries for this platform include extracting a paper's publication date and author names.

**World of Marine Species (WoRMS).** WoRMS[10] serves as the authoritative global database for marine taxonomic information, providing comprehensive data on marine organisms worldwide. The platform features multimedia galleries, species profiles, and taxonomic hierarchies, offering rich visual content alongside traditional text-based scientific information. Our queries include finding an image of a given marine species and the name of the photographer.

---

[5]https://arxiv.org/

[6]https://iclr.cc/

[7]https://icml.cc/

[8]https://neurips.cc/

[9]https://huggingface.co/papers

[10]http://marinespecies.org/

## H.2 AUTOMOTIVE

**Fueleconomy.** Fueleconomy[11] serves as the official U.S. government resource for vehicle fuel economy information, operated jointly by the Environmental Protection Agency (EPA) and the Department of Energy (DOE). The platform provides standardized fuel efficiency data, including Miles Per Gallon (MPG) ratings, and environmental impact information across diverse vehicle categories, ranging from sedans to trucks. Our queries include finding the annual fuel cost for a specific car model and year.

## H.3 SPORTS

**TheSportsDB.** TheSportsDB[12] operates as an open, crowd-sourced sports database providing comprehensive artwork and metadata for sports events, teams, and leagues worldwide. The platform contains detailed information about football clubs, basketball teams, player statistics, match results, and tournament histories across multiple sports disciplines. Our queries include extracting YouTube links for specific matches and retrieving tournament banners for given competitions.

## H.4 LIBRARY

**DPLA.** DPLA[13] serves as the central portal for America's digital cultural heritage, aggregating millions of items from libraries, archives, and museums across the United States. The collection includes historical photographs, manuscripts, books, newspapers, audio recordings, and artifacts spanning American history from colonial times to the present. Queries include extracting images, links, and titles for specific articles or historical documents within the vast digital collection.

## H.5 FOOD

**TheMealDB.** TheMealDB[14] provides a comprehensive open database of meals and recipes from around the world, offering detailed cooking instructions and ingredient information. The database includes thousands of recipes spanning various cuisines, complete with ingredient measurements, cooking steps, and high-quality food photography. Our tasks involve extracting ingredient images and names for specific dishes from the database.

**TheCocktailDB.** TheCocktailDB[15] operates as a complementary database to TheMealDB, specializing in cocktail and drink recipes with detailed preparation instructions. The collection features hundreds of cocktail recipes, including classic drinks, modern mixology creations, and non-alcoholic beverages with ingredient lists, preparation methods, and serving suggestions. Example queries include extracting cocktail images and names for specific drinks from the comprehensive database.

## H.6 BOOK

**Books to Scrape.** Books to Scrape[16] serves as a sandbox website designed specifically for web scraping practice, mimicking the structure and functionality of real online bookstores. The catalog contains 1,000 fictional books across various genres, including mystery, romance, science fiction, and non-fiction, complete with cover images, star ratings, prices, and availability status. Our tasks include extracting book titles, stock status, and pricing information from product catalog structures.

---

[11] https://www.fueleconomy.gov/
[12] https://www.thesportsdb.com/
[13] https://dp.la/
[14] https://www.themealdb.com/
[15] https://www.thecocktaildb.com/
[16] https://books.toscrape.com/

### H.7    E-COMMERCE

**Sandbox Oxylabs.**    Sandbox Oxylabs[17] provides a controlled e-commerce environment specifically designed for web scraping testing, featuring a realistic gaming product marketplace interface. The platform showcases various video games across different platforms, including PC, PlayStation, Xbox, and Nintendo, with detailed product descriptions, screenshots, system requirements, and user reviews. Our queries focus on extracting game titles, genres, and pricing information.

### H.8    OTHERS

**Quotes to Scrape.**    Quotes to Scrape [18] operates as a sandbox website featuring collections of famous quotes with author attribution, designed for web scraping education and practice. The site contains inspirational and philosophical quotes from notable figures throughout history, including writers, philosophers, scientists, and world leaders, organized with tagging systems and author biographical information. Tasks include extracting quotes and their corresponding author names to test basic text parsing capabilities.

**Scrape This Site.**    Scrape This Site [19] provides a comprehensive collection of web scraping challenges with various HTML structures and content types across different pages. The platform offers multiple datasets, including country statistics with population and area data, hockey team information, movie databases with Oscar winners, and sandbox environments for testing different scraping scenarios. Our queries span extracting movie titles and awards, country names, and capitals.

## I    CASE STUDY

We provide successful inference cases of VGS for each task type. The examples for `Type I` to `Type IV` are presented in Figure 26 to Figure 29. For visualization purposes, the inferred bounding boxes are outlined with thick borders.

## J    PROMPT TEMPLATES

We provide the prompt templates used in our experiments. The prompts for VGS are organized by each step: attribute identification (Figure 12), visual grounding (Figure 13), element pinpointing (Figures 14 and 15), and XPath synthesis (Figure 16). For comparison, we also provide the prompts for the baseline methods evaluated in our study. These include the prompts for CoT (Figures 17 and 18), Reflexion (Figures 19, 20, and 21), AutoScraper (Figures 22, 23, and 24), and LLM Extractor (Figure 25).

## K    LIMITATIONS

Our work, while advancing the evaluation of WIE systems, has a few limitations. First, the current version of LIVEWEB-IE is constructed exclusively from English-based websites and natural language queries. Consequently, the performance of WIE systems on our benchmark may not generalize to web pages and user instructions in other languages and cultural contexts. Extending the benchmark to encompass multilingual and multicultural scenarios is a considerable next step toward building universal WIE systems. Second, a limitation pertains to the scope of the benchmark. LIVEWEB-IE is constructed from 15 websites across 8 domains. While this curated selection ensures a stable and high-quality evaluation, the findings may not fully generalize to the vast array of websites, such as highly dynamic social media feeds. Expanding the domain coverage and scale of the benchmark is a crucial step in more rigorously validating generalization performance. Third, attributes and queries are restricted to information with temporally stable values. To ensure reproducible evaluation and maintain stable ground-truth annotations, we curated queries for information

---

[17]https://sandbox.oxylabs.io/

[18]https://quotes.toscrape.com/

[19]https://www.scrapethissite.com/

that is unlikely to change over time. Consequently, while LIVEWEB-IE evaluates a system's robustness to structural drift by accessing live websites, its queries are limited to temporally stable information and do not include dynamic data like current stock prices. While this design aligns with the foundational WIE challenge, analyzing web page structure, and mitigates the overhead of continuous ground-truth maintenance, we acknowledge that expanding the benchmark to include time-sensitive queries is a valuable future direction that would broaden the diversity of tasks.

## L  THE USE OF LARGE LANGUAGE MODELS

Following the ICLR 2026 policies on LLMs usage, we disclose our use of LLMs in the writing process of this paper. The role of LLMs was confined to that of a writing assistant, helping to improve grammatical correctness and readability. It is important to note that the LLM was not used for generating core research ideas and drafting the primary structure of the paper. All model-generated suggestions were critically evaluated, and the final text was written by the authors, who bear full responsibility for the entirety of this work.

Table 5: Results on existing benchmarks.

| Models | Method | SWDE | | | Expanded SWDE | | |
|---|---|---|---|---|---|---|---|
| | | P | R | F1 | P | R | F1 |
| *Proprietary Models* | | | | | | | |
| Gemini 2.5 Flash | COT | 92.62 | 76.82 | 75.33 | 88.47 | 65.08 | 63.74 |
| | Reflexion | 94.31 | 84.53 | 84.40 | 89.50 | 80.14 | 79.22 |
| | AutoScraper | 93.87 | 78.79 | 78.07 | 88.58 | 74.14 | 73.44 |
| | VGS | **96.20** | **86.58** | **88.42** | **95.76** | **81.88** | **83.19** |
| GPT-4o-mini | COT | 88.01 | 73.21 | 69.85 | 87.97 | 71.59 | 68.92 |
| | Reflexion | 92.11 | 68.79 | 68.15 | **94.14** | 68.51 | 68.05 |
| | AutoScraper | 91.67 | 77.94 | 77.00 | 91.46 | **79.88** | **78.62** |
| | VGS | **95.69** | **81.72** | **83.49** | 92.08 | 72.80 | 73.48 |
| GPT-4o | COT | 92.81 | 77.50 | 77.50 | 91.92 | 74.07 | 76.43 |
| | Reflexion | 94.69 | 85.31 | 86.88 | 93.59 | 78.69 | 78.47 |
| | AutoScraper | 94.06 | **88.44** | 87.50 | 91.11 | **84.66** | 81.71 |
| | VGS | **96.92** | 87.56 | **89.77** | **96.07** | 84.25 | **84.19** |
| *Open-Source Models* | | | | | | | |
| Gemma-3-4B | COT | **92.35** | 14.40 | 13.99 | 88.16 | 8.99 | 7.60 |
| | Reflexion | 90.71 | 15.92 | 14.57 | **92.61** | 4.74 | 4.24 |
| | AutoScraper | 86.75 | 24.59 | 23.56 | 87.85 | 15.86 | 14.48 |
| | VGS | 66.62 | **57.43** | **58.86** | 65.32 | **55.28** | **55.67** |
| Gemma-3-27B | COT | 87.12 | 49.89 | 47.24 | 82.99 | 43.84 | 39.01 |
| | Reflexion | 88.22 | 54.27 | 52.02 | **88.94** | 49.15 | 46.29 |
| | AutoScraper | **88.90** | 58.65 | 57.67 | 88.24 | 59.02 | 55.28 |
| | VGS | 77.14 | **67.07** | **68.28** | 76.06 | **62.53** | **62.97** |
| Qwen-2.5-7B | COT | 91.91 | 33.91 | 33.36 | 91.21 | 28.20 | 26.68 |
| | Reflexion | **93.33** | 39.46 | 39.27 | 91.28 | 35.30 | 34.49 |
| | AutoScraper | 92.68 | 32.81 | 32.50 | **94.87** | 30.95 | 30.60 |
| | VGS | 71.39 | **60.16** | **61.88** | 73.42 | **60.55** | **61.75** |
| Qwen-2.5-32B | COT | 89.65 | 69.55 | 67.91 | 89.10 | 67.87 | 65.08 |
| | Reflexion | 95.48 | 71.82 | 71.35 | 94.29 | 67.44 | 67.42 |
| | AutoScraper | 93.94 | 69.03 | 68.03 | 91.82 | 64.67 | 63.56 |
| | VGS | **97.04** | **77.87** | **80.29** | **95.92** | **74.98** | **77.07** |
| Qwen-2.5-72B | COT | 92.83 | 71.91 | 70.54 | 89.23 | 61.00 | 58.76 |
| | Reflexion | **94.95** | 73.38 | 73.85 | 91.70 | 65.89 | 65.33 |
| | AutoScraper | 92.78 | **83.51** | 83.02 | 89.85 | 78.63 | 77.83 |
| | VGS | 94.38 | 81.65 | **83.66** | **92.95** | **79.75** | **80.74** |

Table 6: Statistics of SWDE-sub, a manually curated subset where target values remain identical to recent web pages.

| Domain | Website | Attributes | # web pages |
|---|---|---|---|
| NBAPlayer | espn | name | 10 |
| | nba | name | 10 |
| | usatoday | height, name | 10 |
| | yahoo | height, name, weight | 10 |
| | wiki | height, name, weight | 10 |
| Movie | allmovie | director, title | 10 |
| | boxofficemojo | title | 10 |
| | imdb | director, title | 10 |
| | metacritic | director, title | 10 |
| | rottentomatoes | director, title | 10 |

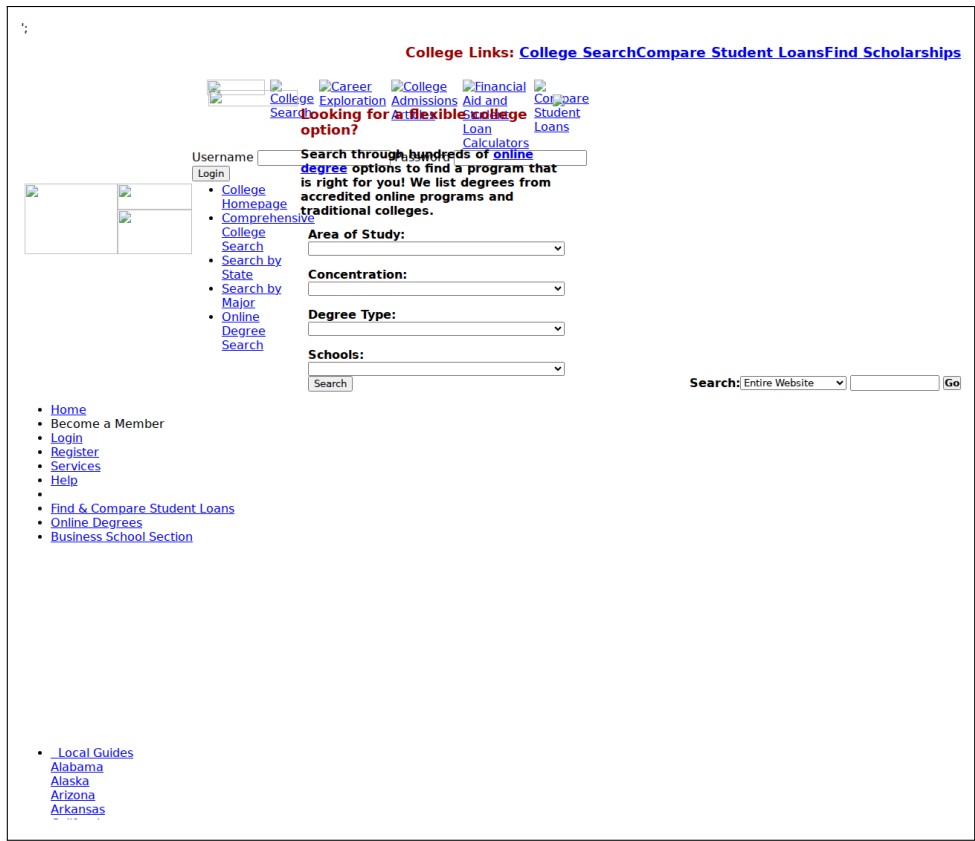

Figure 11: An example of a web page excluded due to text overlap.

Table 7: Dataset statistic (Part 1): Academic and Auto domains.

| Domain | Website | Attribute | # web page | # Group |
|---|---|---|---|---|
| Academic | ArXiv | author
author profile link
pdf link
subject
title | 87 | 3 |
| | ICLR | abstract link
author
author profile link
title | 292 | 2 |
| | NeurIPS | abstract link
author
author profile link
title | 311 | 2 |
| | ICML | abstract link
author
author profile link
title | 225 | 2 |
| | Hugging Face | author
author count
citing model link
model citation count
paper link
publish date
source link
title | 230 | 2 |
| | WoRMS | author
author profile link
country
editor profile link
editor type
group
group image
group link
institute
photo title
species
species image
species link | 73 | 4 |
| Auto | Fueleconomy | annual fuel cost
driver mpg
tank size
vehicle
vehicle image
vehicle link
vehicle type | 69 | 3 |

Table 8: Dataset statistic (Part 2): Sports, Library, and Food domains.

| Domain | Website | Attribute | # web page | # Group |
|--------|---------|-----------|------------|---------|
| Sports | TheSportsDB | banner
event link
fanart
league date
league link
lineup
logo
member image
member link
poster
profile link
search link
season badge
season poster
team
team badge
team link
thumb image
youtube link | 261 | 6 |
| Library | DPLA | article image
article link
collection
collection image
collection link
exhibition link
source image
source link
source title
subject link
title
topic
topic link | 184 | 5 |
| Food | TheMealDB | flag image
ingredient
ingredient image
ingredient link
meal
meal image
meal link | 298 | 2 |
| | TheCocktailDB | cocktail
cocktail image
cocktail link
ingredient
ingredient image
ingredient link | 495 | 2 |

Table 9: Dataset statistic (Part 3): Book, E-commerce, and Others domains.

| Domain | Website | Attribute | # web page | # Group |
|---|---|---|---|---|
| Book | Books to Scrape | book image
book link
category
count
price
stock status
title
upc | 1100 | 3 |
| E-commerce | Sandbox Oxylabs | developer
game count
genre
image
link
platform
price
title | 1143 | 3 |
| Others | Scrape This Site | area
awards
capital
country
goals for
nominations
population
team
title
turtle image
turtle link
turtle name
wins
year | 32 | 4 |
| | Quotes to Scrape | author
author profile link
birth
quote
tag link | 70 | 3 |

---

**VGS Prompt for Attribute Identification**

```
You are an Attribute Extractor that converts user extraction requests into structured JSON payloads
 for web scraping systems.

Task: Transform a user's data extraction request into a compact JSON specification that defines
what attributes to extract.

Instructions:
- Analyze the user's request to identify specific data points they want extracted
- Focus on the minimal set of attributes needed to fulfill the user's request

Rules:
- JSON Structure: Return a JSON object with exactly one key: "attributes"
- The "attributes" key must contain a list of attribute names
- Minimal Scope: Include only attributes explicitly requested or clearly implied by the user's task
- Output Only: Return the JSON object with no code fences, explanations, or additional text

Please output in the following JSON format:
{
  "attributes": [
    "attribute_name_1",
    "attribute_name_2"
  ]
}
```

Figure 12: VGS prompt template for attribute identification.

---

**VGS Prompt for Visual Grounding**

```
You are a visual grounding assistant that identifies which webpage contains a specific UI element
from a set of viewport screenshots.

Task: Given a set of viewport screenshots and a requested attribute, determine which specific
region (if any) contains the target attribute clearly visible within its boundaries.

Instructions:
- You will receive multiple viewport screenshots representing different regions of a webpage
- Each screenshot will be labeled with a region identifier
- Analyze each region independently to determine if it contains the target attribute

Rules:
- Evaluate each region screenshot individually for the presence of the target attribute
- If multiple regions contain the target attribute, select the one where it is most clearly visible
 and prominent
- Do not make assumptions about page structure or infer elements beyond what is directly observable
- Do not assume elements exist based on typical webpage patterns or templates
- Focus only on elements that are clearly within each region's viewport boundaries
- Return only the single best-matching region

Please output in the following JSON format:
{
  "matching_region": "" # Single region ID that best contains the target attribute
}
```

Figure 13: VGS prompt template for visual grounding.

---

**VGS prompt for Element Pinpointing (Element Scanning)**

You are a precise visual extractor that analyzes a web UI screenshot to extract specific attribute content based on modality classification.

Task: Given a viewport screenshot and an attribute name, determine the correct modality from the attribute name and perform the appropriate extraction task.

Instructions:
Classify the attribute into exactly one modality based on naming patterns:
1. TEXT Modality – For textual attributes
   – Extract all visible text strings exactly as they appear
   – Preserve natural reading order (top-to-bottom, left-to-right)
   – No text normalization or paraphrasing
   – "tags" key MUST be empty

2. HYPERLINK Modality – For link related attributes
   – Output HTML 'a' tags only
   – Ignore visible link text content
   – "texts" key MUST be empty

3. IMAGE Modality – For visual content attributes
   – Output 'img' tags only
   – Ignore any text overlays on images
   – "texts" key MUST be empty

Rules:
– Select exactly one modality – never mix text and tag outputs
– For TEXT modality: deduplicate while preserving reading order
– For HYPERLINK and IMAGE modalities: use lowercase HTML tag names only, keep minimal and generic

Please output in the following JSON format:
{
  "texts": [], # For TEXT modality: visible text strings in reading order, exactly as they appear
  "tags": [] # For non-TEXT modalities: lowercase HTML tag names
}

Figure 14: VGS prompt template for element pinpointing (element scanning).

---

**VGS prompt for Element Pinpointing (Element Selection)**

You are a precise visual element selector that analyzes an annotated region screenshot to identify specific UI elements matching a target attribute.

Task: Given an annotated region screenshot where potential candidate elements have colored bounding boxes with unique numerical labels, select the exact bounding box IDs that contain values for the specified target attribute.

Instructions:
– Each element is outlined with a colored box containing a small ID label in the top-right corner
– The label shows an integer ID with white text on a colored background matching the box color
– Focus on elements that directly contain or represent the values of the target attribute

Rules:
– Relevance Only: Return IDs only for bounding boxes that directly contain values of the target attribute
– Leaf-Level Selection: When nested boxes exist, prefer the innermost box containing the raw data
– Complete Coverage: Return all matching leaf elements visible on screen, not just the first occurrence
– No Ambiguous Containers: Avoid selecting wrapper divs, sections, or containers when inner data elements exist
– Whitelist Only: Only output IDs that exist in the screenshot; do not invent or guess numbers

Please output only the integer IDs as a simple list:
[]

Figure 15: VGS prompt template for element pinpointing (element selection).

---

**VGS Prompt for Xpath Synthesis**

You are an expert XPath generator that creates robust, generalizable selectors for web scraping based on HTML segments, marked region, and target attribute.

Task: Generate one reliable XPath selector from the provided HTML segments, marked region screenshot, and target attribute that will work across structurally similar pages.

Instructions:
- Analyze HTML segments and marked regions to identify structural patterns and stable anchoring elements
- Use the marked region screenshot as visual context to understand the target element's location and appearance

Rules:
- Pattern Recognition: Identify common structural patterns across all HTML segments
- Context Integration: Leverage both visual evidence and localized HTML information
- Generalizability: Ensure XPath works reliably across similar page structures
- Ignore irrelevant or noisy HTML samples when forming the generalized selector

Please output in the following JSON format:
```
{
  "xpath": "" # Single XPath selector targeting the appropriate content
}
```

Figure 16: VGS prompt template for Xpath synthesis.

---

**CoT Prompt for Top-Down Operation**

You are a web parser that is good at reading and understanding the HTML code and can give clear executable code on the browser.

Please read the following HTML code, and then return XPaths grouped by field names that directly match the elements in the page, satisfying the instructions below.

Instruction: {0}

Rules:
- Field Determination:
  - Use explicitly listed field names from the instruction
  - Otherwise, infer minimal relevant fields present in the DOM without inventing unsupported fields
- XPath Construction:
  - Avoid embedding exact literal values or visible text from the HTML content
  - Create structurally robust selectors using attribute and structural patterns
  - Prefer stable element attributes and hierarchical relationships over brittle identifiers
- Field Organization:
  - Generate separate XPaths for each target node when multiple nodes exist for a field
  - Maintain distinct XPaths that satisfy the instruction requirements for each field
- Data Structure Requirements:
  - Ensure xpath and value dictionaries have identical key sets
  - Align the XPath and value lists so each position corresponds to the same target node
  - Extract text from single nodes without concatenating content from multiple elements
- Missing Information Handling:
  - Return empty lists for explicitly enumerated fields when HTML lacks suitable content
  - Return empty objects for both xpath and value when no instruction fields are specified and no suitable content is found

Please output in the following JSON format:
```
{
  "thought": "", # Brief reasoning on field selection and XPath derivation
  "value": { # Field-to-string-list mapping of extracted text content
    {} # dynamic keys
  },
  "xpath": { # Field-to-XPath-list mapping for target node selection
    {} # dynamic keys
  }
}
```

Here's the HTML code:
```
{1}
```

Figure 17: CoT prompt template for top-down operation.

---

**CoT Prompt for Synthesis Operation**

```
You are a perfect discriminator, which is good at HTML understanding as well.

Following the instruction, there are some action sequence written from several HTML and the
corresponding result extracted from several HTML. Please choose one that can potentially be best
adapted to the same extraction task on other webpages on the same website.

Here are the instructions for the task:
Instructions: {0}

The action sequences and the corresponding extracted results with different sequences on different
webpages are as follows:
{1}

Please output in the following JSON format:
{
    "thought": "" # brief thinking about which to choose
    "number": "" # the best action sequence chosen from the candidates, starts from 0.
}
```

Figure 18: CoT prompt template for synthesis operation.

---

**Reflexion Prompt for Top-Down Operation**

```
You are a web parser that is good at reading and understanding the HTML code and can give clear
executable code on the browser.

Please read the following HTML code, and then return XPaths grouped by field names that directly
match the elements in the page, satisfying the instructions below.

Instruction: {0}

Rules:
- Field Determination:
  - Use explicitly listed field names from the instruction
  - Otherwise, infer minimal relevant fields present in the DOM without inventing unsupported fields
- XPath Construction:
  - Avoid embedding exact literal values or visible text from the HTML content
  - Create structurally robust selectors using attribute and structural patterns
  - Prefer stable element attributes and hierarchical relationships over brittle identifiers
- Field Organization:
  - Generate separate XPaths for each target node when multiple nodes exist for a field
  - Maintain distinct XPaths that satisfy the instruction requirements for each field
- Data Structure Requirements:
  - Ensure xpath and value dictionaries have identical key sets
  - Align the XPath and value lists so each position corresponds to the same target node
  - Extract text from single nodes without concatenating content from multiple elements
- Missing Information Handling:
  - Return empty lists for explicitly enumerated fields when HTML lacks suitable content
  - Return empty objects for both xpath and value when no instruction fields are specified and no
suitable content is found

Please output in the following JSON format:

{
  "thought": "", # Brief reasoning on field selection and XPath derivation
  "value": { # Field-to-string-list mapping of extracted text content
    {} # dynamic keys
  },
  "xpath": { # Field-to-XPath-list mapping for target node selection
    {} # dynamic keys
  }
}

Here's the HTML code:
```
{1}
```
```

Figure 19: Reflexion prompt template for top-down operation.

---

**Reflexion Prompt for Self-Reflection Operation**

```
Please read the following HTML code, and then return all possible XPaths that can recognize the
elements in the HTML matching the instructions below.
Instruction: {0}

Rules:
- History Analysis:
  - Evaluate consistency between extraction results and expected values
  - Identify irrelevant elements that were incorrectly captured
  - Check for empty results indicating failed extraction
  - Accept raw values with redundant separators as consistent, since post-processing will handle
them
- Value Assessment:
  - Re-examine expected values in the context of the HTML structure
  - Determine optimal XPath strategies for locating target content
  - Consider structural patterns and element relationships for reliable targeting
- XPath Refinement:
  - Generate new XPaths when current ones fail to meet requirements
  - Retain existing XPaths when they demonstrate accurate extraction
  - Base decisions on analysis findings and extraction quality assessment
- XPath Construction Constraints:
  - Avoid embedding exact literal values or specific HTML element content
  - Prevent overly broad selectors that match multiple nodes with different meanings
  - Use specific class attributes and positional indicators to differentiate target nodes
  - Maintain precision through structural anchors and attribute-based targeting
- Missing Content Handling:
  - Return empty outputs when HTML lacks information matching the instruction
  - Acknowledge extraction limitations rather than forcing inappropriate matches

Please output in the following JSON format:
{
    "thought": "", # Brief reasoning on field selection and XPath derivation
    "consistent": "", # whether the extracted results are consistent with the expected values,
return yes/no directly
    "value": { # Field-to-string-list mapping of extracted text content
      {} # dynamic keys
    },
    "xpath": { # Field-to-XPath-list mapping for target node selection
      {} # dynamic keys
    }
}

And here's the history about the thoughts, XPaths, and results extracted by the crawler:
{1}

Here's the HTML code:
```
{2}
```
```

Figure 20: Reflexion prompt template for self-reflection operation.

---

**Reflexion Prompt for Synthesis Operation**

```
You are a perfect discriminator, which is good at HTML understanding as well.

Following the instruction, there are some action sequence written from several HTML and the
corresponding result extracted from several HTML. Please choose one that can potentially be best
adapted to the same extraction task on other webpages on the same website.

Here are the instructions for the task:
Instructions: {0}

The action sequences and the corresponding extracted results with different sequences on different
webpages are as follows:
{1}

Please output in the following JSON format:
{
    "thought": "" # brief thinking about which to choose
    "number": "" # the best action sequence chosen from the candidates, starts from 0.
}
```

Figure 21: Reflexion prompt template for synthesis operation.

**AutoScraper Prompt for Top-Down Operation**

```
You are a web parser that is good at reading and understanding the HTML code and can give clear
executable code on the browser.

Please read the following HTML code, and then return XPaths grouped by field names that directly
match the elements in the page, satisfying the instructions below.

Instruction: {0}

Rules:
- Field Determination:
  - Use explicitly listed field names from the instruction
  - Otherwise, infer minimal relevant fields present in the DOM without inventing unsupported fields
- XPath Construction:
  - Avoid embedding exact literal values or visible text from the HTML content
  - Create structurally robust selectors using attribute and structural patterns
  - Prefer stable element attributes and hierarchical relationships over brittle identifiers
- Field Organization:
  - Generate separate XPaths for each target node when multiple nodes exist for a field
  - Maintain distinct XPaths that satisfy the instruction requirements for each field
- Data Structure Requirements:
  - Ensure xpath and value dictionaries have identical key sets
  - Align the XPath and value lists so each position corresponds to the same target node
  - Extract text from single nodes without concatenating content from multiple elements
- Missing Information Handling:
  - Return empty lists for explicitly enumerated fields when HTML lacks suitable content
  - Return empty objects for both xpath and value when no instruction fields are specified and no
suitable content is found

Please output in the following JSON format:

{
  "thought": "", # Brief reasoning on field selection and XPath derivation
  "value": { # Field-to-string-list mapping of extracted text content
    {} # dynamic keys
  },
  "xpath": { # Field-to-XPath-list mapping for target node selection
    {} # dynamic keys
  }
}

Here's the HTML code:
```
{1}
```
```

Figure 22: AutoScraper prompt template for top-down operation.

**AutoScraper Prompt for Step-back Operation**

```
Your main task is to judge whether the following HTML code contains all the expected values, which
are recognized beforehand.
Instruction: {0}
And here's the value (note: there will be at most 10 expected values): {1}
The HTML code is as follows:
```
{2}
```

Please output in the following JSON format:
{
    "thought": "", # a brief thinking about whether the HTML code contains the expected value
    "judgement": "" # whether the HTML code contains all extracted values. Return yes/no directly.
}
```

Figure 23: AutoScraper prompt template for step-back operation.

---

**AutoScraper Prompt for Synthesis Operation**

```
You are a perfect discriminator, which is good at HTML understanding as well.

Following the instruction, there are some action sequence written from several HTML and the
corresponding result extracted from several HTML. Please choose one that can potentially be best
adapted to the same extraction task on other webpages on the same website.

Here are the instructions for the task:
Instructions: {0}

The action sequences and the corresponding extracted results with different sequences on different
webpages are as follows:
{1}

Please output in the following JSON format:
{
    "thought": "" # brief thinking about which to choose
    "number": "" # the best action sequence chosen from the candidates, starts from 0. If there is
none, output 0.
}
```

Figure 24: AutoScraper prompt template for synthesis operation.

---

**LLM Extractor Prompt**

```
You are an AI Extractor that performs STRICT single-page structured extraction.

Rules:
- Infer attribute names from the user instruction in natural language
- Return empty lists for attributes with no values
- Output ONLY the final JSON object with NO additional text or explanations

Here are the instructions of the task: {0}
HTML: {1}

Please output in the following JSON format:
{
    "<attribute>": ["value1", "value2"] # List of extracted values for this attribute
}
```

Figure 25: LLM Extractor prompt template.

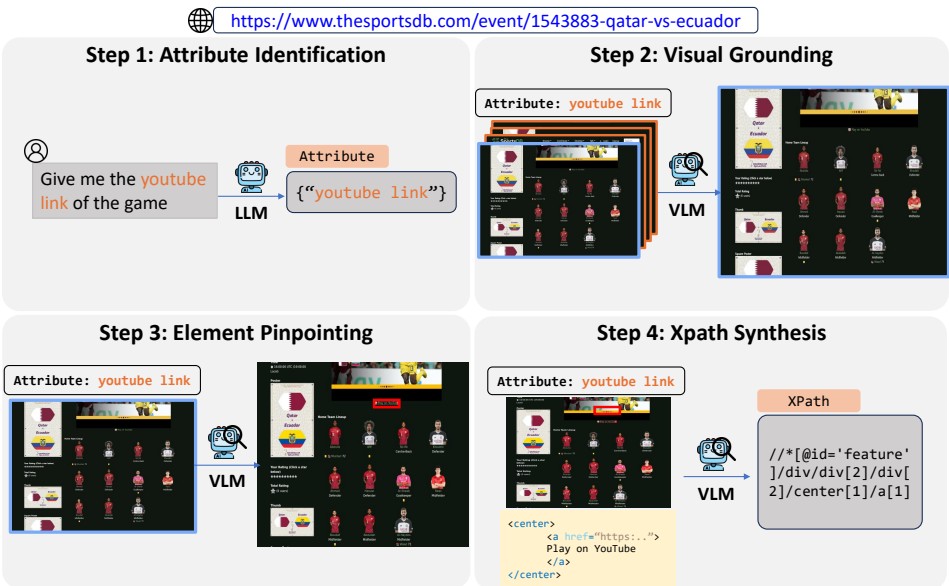

Figure 26: Complete VGS trajectory for Type I task on LIVEWEB-IE.

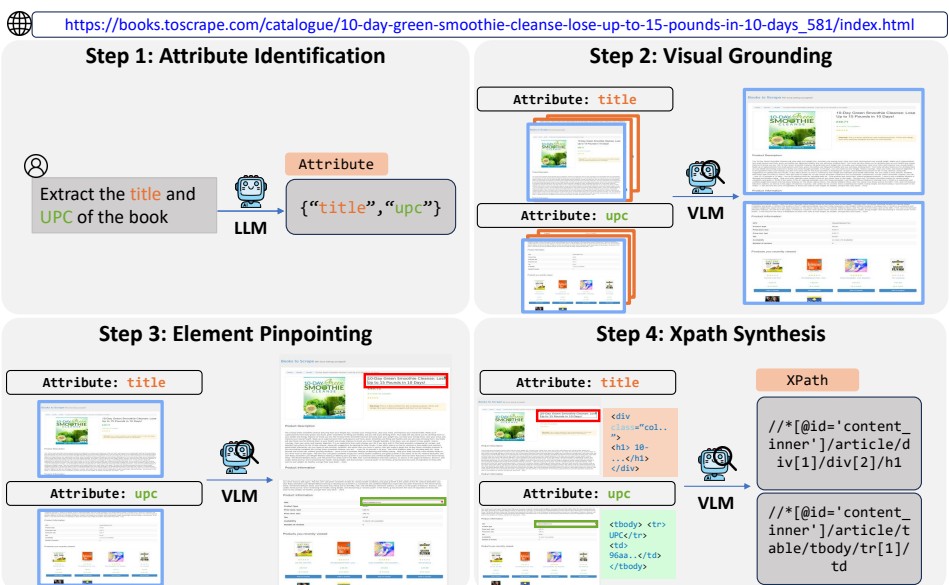

Figure 27: Complete VGS trajectory for Type II task on LIVEWEB-IE.

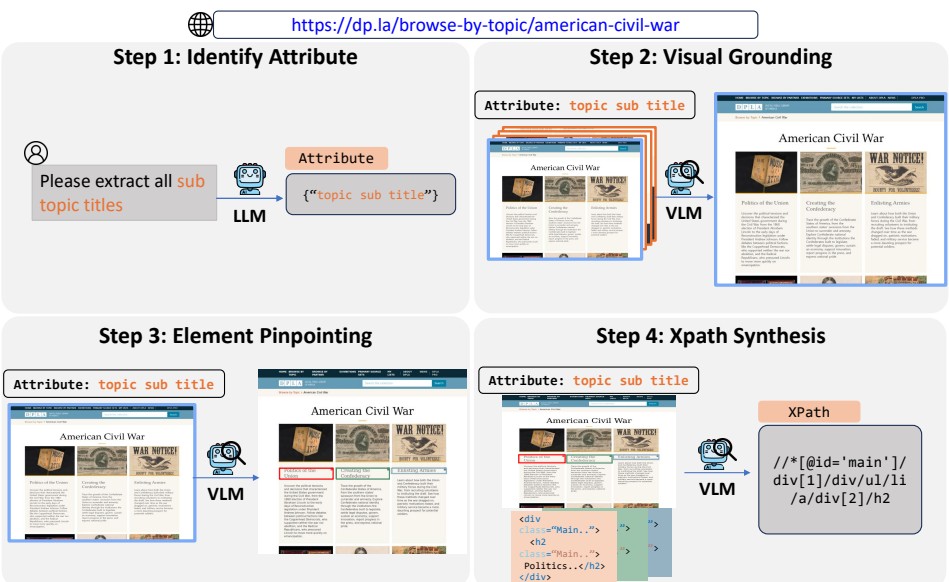

Figure 28: Complete VGS trajectory for `Type III` task on LIVEWEB-IE.

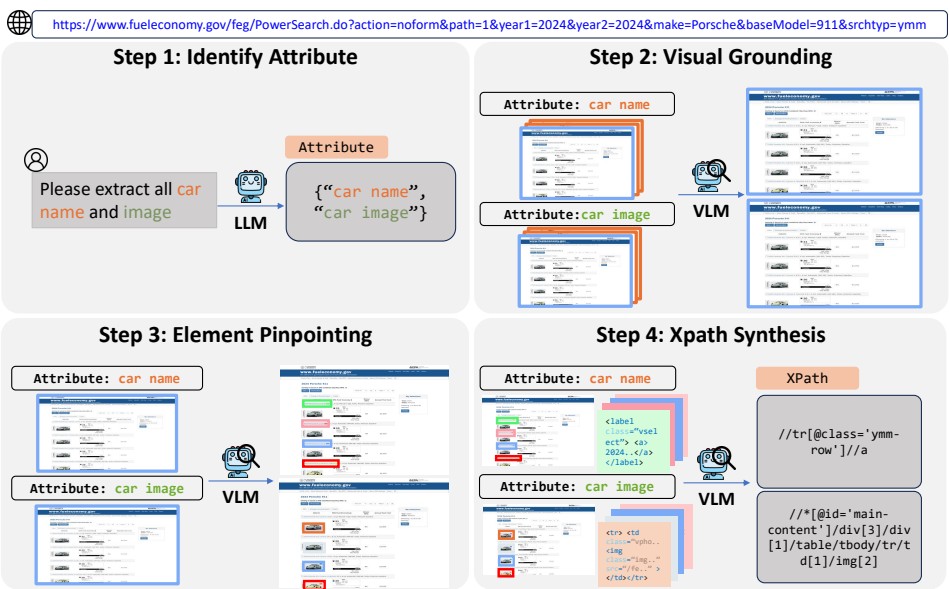

Figure 29: Complete VGS trajectory for `Type IV` task on LIVEWEB-IE.

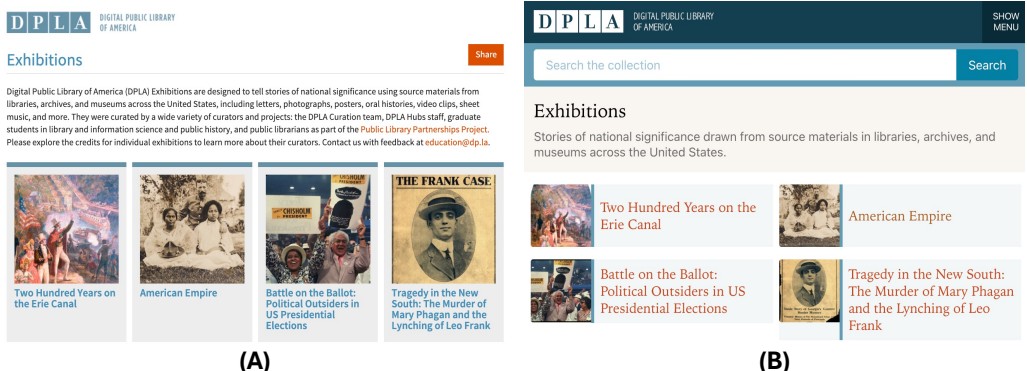

Figure 30: Comparison of layout changes on the DPLA website over time. **(A)** illustrates the web page from a past version (January 2018), which was retrieved from the Web Archive (`https://web.archive.org`), a digital archive preserving over 1 trillion historical web page snapshots, while **(B)** depicts the current version (November 2025) of the same URL. The attributes "title", "article image", and "article link" retain identical values across both versions, but the visual layout and underlying DOM structure of the page have diverged. By targeting attributes whose information remains consistent regardless of layout changes, we enable a stable evaluation of a WIE system's information extraction performance on web pages as they exist at the current moment.

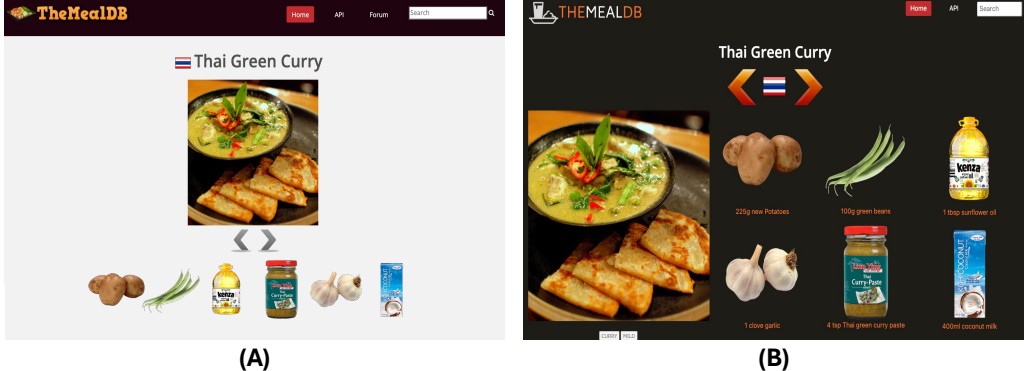

Figure 31: Comparison of layout changes on the TheMealDB website over time. **(A)** illustrates a specific meal page from a past version (October 2017), while **(B)** depicts the current version (November 2025) of the same URL. The attributes "flag", "ingredient image", "ingredient link", "meal image", and "meal" retain identical values across both versions, but the visual layout and underlying DOM structure of the page have diverged.

