# OpenReview forum: "LiveWeb-IE: A Benchmark For Online Web Information Extraction"
_ICLR.cc/2026/Conference — ICLR 2026 Poster_

### Official Review · Reviewer_D9Lx · 2025-10-20

**Soundness:** 2
**Presentation:** 2
**Contribution:** 2
**Rating:** 4
**Confidence:** 3

**Summary:**

The paper presents LIVEWEB‑IE, a benchmark for evaluating web information extraction systems on so‑called live websites, and introduces a baseline model, Visual Grounding Scraper (VGS). The benchmark is constructed through website selection, annotation, and verification, emphasizing content stability for reproducibility. However, since it deliberately focuses on static webpages and stable answers, the setting does not clearly demonstrate advantages over conventional snapshot‑based benchmarks in reflecting real temporal dynamics.

**Strengths:**

The benchmark clearly defines four task types covering single/multi‑attribute and single/multi‑value extraction, providing a systematic and comprehensive framing of the web information extraction problem.

The paper presents a coherent pipeline for website selection, annotation, and verification, showing solid engineering effort.

The introduction of the baseline model (Visual Grounding Scraper – VGS) illustrates the benchmark’s intended use and offers an initial reference for comparison.

**Weaknesses:**

Although the paper highlights the live nature of the benchmark, most of the evaluated webpages are essentially static and rarely updated. As a result, the work does not clearly demonstrate any advantage of using “live” data over conventional snapshot‑based settings that are already stable and reproducible. Even HotpotQA, which is built on fixed Wikipedia content and unchanging commonsense facts, offers a comparable level of variability.

The benchmark explicitly filters for stable pages and attributes “unlikely to change,” ensuring reproducibility but sidestepping what genuinely makes web extraction live—coping with temporal and structural drift. A truly live evaluation should treat the target as a function responsive to time and environment rather than a fixed, time‑agnostic value.

Incorporating multiple modalities in web information extraction is intuitively reasonable, but the paper does not clearly explain how these modalities are functionally integrated or why they are critical to the benchmark’s objectives. The description and experiments leave it unclear whether multimodality influences task formulation, model behavior, or evaluation outcomes, making this aspect feel under‑motivated despite being prominently emphasized.

**Questions:**

Could the authors clarify what specific phenomena of “liveness” are actually captured in the current setup, beyond what a static snapshot benchmark would provide?

Given that the dataset construction deliberately filters for stable content, how does the current benchmark evaluate a system’s robustness to temporal or structural changes—if at all?

Multimodality is highlighted as a major aspect of the work, but its functional role is unclear. Can the authors specify which tasks or results concretely rely on non‑textual modalities, and what insights would be lost if these were removed?

---

> ### Author Response · Authors · 2025-11-20
>
> *We thank the reviewer for insightful comments, constructive feedback, and raising important questions. We hope our responses and the accompanying revisions fully address the reviewer's concerns.*
>
> ### **D9Lx-W1&Q1**
> >The paper does not clearly demonstrate the advantage of using live data over static snapshots; clarification on the specific captured liveness phenomena is required.
>
> We would like to clarify a distinction at the core of our design: the stability we filter for applies only to the ground-truth **values**, not the **layout**. This ensures evaluation validity by focusing on stable values, which is a separate consideration from the dynamic nature of a web page's layout. **We kindly ask the reviewer to refer to the newly added Fig. 30 and Fig. 31**, which we included to explicitly illustrate this distinction.
>
> Conventional HTML snapshots, captured years in the past, fail to represent the current web structures. Given that WIE performance is strongly dependent on the structural properties of web pages, this may lead to a generalization gap (**Appendix F, Tab.4**). The "liveness" of LiveWeb-IE is in its evaluation protocol: it mandates evaluation directly against live URLs, forcing WIE systems to process the current web page structure at the moment of testing (**§ 3.1**).
>
> To ensure a valid benchmark that captures structural drift, we selected stable, fact-based ground truths. For example, even if a web page redesigns its layout, the ground-truth value for a query like "2022 World Cup final result" remains constant.
>
> In summary, evaluating systems against live URLs allows us to capture the "liveness" of a web page's structural drift, a phenomenon that static snapshots fail to capture. This addresses the core point: our claim of "stability" refers only to the fact-based ground-truth values, which ensure evaluation validity, while the layout itself remains dynamic and subject to evolution [1].
>
> To further clarify our benchmark's purpose and design, **we added a more detailed explanation to § 1 and § 3.**
>
>
> ### **D9Lx-W2**
> >Benchmark design avoids genuine liveness by filtering for stable pages, sidestepping temporal and structural drift.
>
> We fully agree with the reviewer that evaluating time-responsive targets (e.g., stock tickers, weather updates) represents a valuable and distinct dimension of "liveness." We acknowledge this as a meaningful future direction to broaden task diversity and **expand our Limitations section to discuss this scope.**
>
> However, our decision to prioritize structural drift over value changes is strategic, as it aligns with the fundamental, long-standing challenge of the WIE task. The core objective of WIE is to identify the structural location of a value within the web page (e.g., finding the XPath for "price"), which is independent of what that value is (e.g., whether the price is $10 or $20):
> - Traditional WIE research has been dominated by methods highly sensitive to structural properties, such as wrapper-induction systems that rely on DOM patterns [2,3].
> - Modern approaches that generate reusable wrappers are based on analyzing and understanding the HTML structure to create robust XPaths or selectors [4,5].
>
> By focusing on stable values within dynamic layouts, we can accurately measure how well a WIE system handles the web structure at the time of testing. A further benefit of this design is that it ensures the benchmark remains reproducible over time without needing constant manual updates. **We added a design rationale and its advantages to § 3.2.**
>
>
> [1] Mialon et al. GAIA: A Benchmark for General AI Assistants, arXiv'23.11.
> [2] Crescenzi et al. RoadRunner: Towards Automatic Data Extraction from Large Web Sites, VLDB 2001.
> [3] Lerman et al. Wrapper maintenance: A machine learning approach, JAIR 2003.
> [4] Huang et al. AutoScraper: A progressive understanding web agent for web scraper generation, EMNLP 2024.
> [5] Kaur et al. Automating xpath query generation using nlp for streamlined web crawling and gui testing, ICTEST 2025.

---

> ### Author Response · Authors · 2025-11-20
>
> ### **D9Lx-W3&Q3**
> >The functional role of multimodality is unclear; clarification is needed on its integration, criticality, and concrete reliance in tasks or results.
>
> The inclusion of multiple modalities is motivated by the goal of our benchmark, which is to evaluate WIE in real-world scenarios. On the live web, user queries are not limited to text; users often want to extract non-textual information, such as "get the thumbnail for this product," "find all links to author profiles," or "fetch the fantart." A benchmark that ignores these modalities would fail to capture a significant portion of real-world WIE tasks.
>
> This principle is functionally integrated into the benchmark's design and annotation. Our data construction process, detailed in **Appendix C**, required annotators to identify attributes encompassing "not only text but also images and hyperlinks". For non-textual attributes, their source values were collected as the ground truth following the HTML Standard, such as @src for images and @href for hyperlinks.
>
> Furthermore, evaluating these non-textual modalities provides specific insights into model performance. Our analysis (**§ 6.2**) reveals that all methods exhibit a performance degradation when extracting non-textual data. This finding highlights that non-textual extraction remains a key challenge for current systems. This insight is particularly valuable as it directly validates our benchmark's design; without explicitly constructing the dataset to include these modalities, this critical weakness in modern WIE systems would remain undetected.
>
> For a concrete example of a task that requires extracting non-textual modalities, we kindly ask the reviewer to refer to **Fig. 3**, which illustrates a scenario within our benchmark that requires extracting an image (specifically, the 'fanart' of a game).
>
> **We added clarifications to the Introduction (§ 1), Dataset (§ 3.2), and Discussion (§ 6.2)** sections to make the motivation and functional integration of multimodality in our benchmark more explicit.
>
> ### **D9Lx-Q2**
> >How does the benchmark evaluate robustness to temporal or structural changes, given the deliberate filtering for stable content?
>
> Our benchmark evaluates a system's robustness to structural changes precisely because our evaluation protocol mandates that systems access the live URL at the moment of evaluation. Therefore, if a webpage's layout has evolved since our initial verification, the WIE system is immediately confronted with this new layout. A system's robustness to structural change can be explicitly measured by comparing its performance on the LiveWeb-IE benchmark at two different points in time (e.g., its score in November 2025 vs. its score in May 2026). A performance drop would quantify the system's failure to generalize to the structural evolution.

---

> ### Author Response · Authors · 2025-11-26
> **A gentle reminder to reviewer D9Lx**
>
> Dear reviewer D9Lx,
>
> We appreciate your insightful comments regarding the benchmark's design. Following your feedback, we have clarified our focus on structural layout changes while acknowledging time-responsive targets as a future direction. Additionally, we have explicitly detailed the role of multimodality to justify its inclusion in the benchmark. We hope our responses and the newly added contents for these points align with your expectations.
>
> Best regards,
> Paper 11748 Authors

---

> > ### Comment · Reviewer_D9Lx · 2025-11-27
> >
> > I thank the authors for the detailed rebuttal and the clarifications.
> >
> > The clarification that “stability” applies only to fact-based ground-truth values, while the page layout and DOM remain live, is helpful. It makes the intended scope—focusing on structural robustness rather than time-varying values—more explicit. I also find the motivation and implementation of multimodality clearer now: including image and link attributes, and showing consistent performance degradation on these non-textual fields, does give a concrete functional role to multimodality in the benchmark.
> >
> > That said, my main concern about the claimed contribution of “live” remains. Robustness to variation in layout and phrasing is a fundamental requirement for WIE models, and a well-designed static benchmark (with diverse sites/templates, multiple versions, or explicit structural perturbations) can also test this, without requiring live evaluation. In other words, the need to generalize across different layouts and textual realizations does not arise uniquely because the pages are live.
> >
> > Since the benchmark filters out dynamic targets and keeps only fact-like, stable fields, the main effect of being “live” is to introduce layout/structural drift. But robustness to layout variation can also be probed with a sufficiently diverse static benchmark (multiple sites/templates, versions, or perturbations). Under this design, I do not see a clear, demonstrated advantage of “live” over well-constructed static alternatives, even though it is presented as the main point of the paper.
> >
> > The rebuttal improves the framing and strengthens the case for the multimodal aspect, but it does not fully resolve my core concern about the necessity and demonstrated benefit of liveness over static alternatives.

---

> ### Author Response · Authors · 2025-11-27
> **Follow-up response to reviewer D9Lx**
>
> We thank the reviewer for the thoughtful feedback. We appreciate that the reviewer found our clarification on "stability" helpful and recognized the functional value of "multimodality" in our benchmark.
>
> Regarding the capability of static benchmarks, we agree with the reviewer that a well-constructed static benchmark can measure robustness to layout variations. However, as emphasized in **§1** and **§3**, our primary goal is to ensure the **accurate measurement of WIE system performance at the time of evaluation.** We perceive a critical distinction between ensuring robustness to layout diversity within a fixed era and ensuring adaptability to the web's structural evolution. While static benchmarks can address the former by aggregating diverse templates captured at a specific point in time, they fundamentally fail at the latter because they cannot represent the new structural paradigms (e.g., Custom Elements) that emerge after data collection. Since WIE performance is strongly dependent on the structural properties of web pages, evaluating systems on outdated snapshots—no matter how diverse—fails to reflect their true capability in the current web environment.
>
> We support the necessity of the "live" through two key dimensions:
> 1. Continuous Structural Shift
> The structural norms of the web evolve continuously. Modern web pages have shifted from simple, semantic HTML to complex DOM trees constructed via JavaScript frameworks (e.g., React, Vue) and Custom Elements [6]. A fixed benchmark constructed in the past cannot anticipate or represent these modern structural patterns. Therefore, relying on static datasets restricts evaluation to obsolete paradigms and fails to capture the modern web structure.
>
> 2. Evidence of the Generalization Gap
> In **Appendix F**, we empirically validated this gap by comparing performance on offline snapshots (SWDE) against their current live versions, strictly controlling for identical ground-truth values to isolate the effect of structural drift. The results confirmed that existing methods suffer significant performance degradation when applied to the live web; specifically, HTML-based baselines showed an average F1 drop of over 15% when using GPT-4o. This gap demonstrates that static benchmarks fail to predict actual system performance in the modern web environment.
>
> To sum up, while static benchmarks can assess adaptability to diverse layouts from a fixed point in time, they cannot account for the structural drift caused by the web's evolution. Consequently, "Live" evaluation is the protocol that bridges this generalization gap, thereby enabling the accurate measurement of a WIE system's performance at the exact time of evaluation.
>
>
> ### **Summary: Clarification on Benchmark Design and Liveness**
> LiveWeb-IE is designed to evaluate WIE systems directly against live websites, targeting factually stable values to ensure reproducibility while mandating systems to process the web page structure at the moment of evaluation. This design aligns with the fundamental, long-standing challenge of the WIE task and ensures the benchmark remains reproducible over time without needing constant manual updates. While well-constructed static benchmarks can assess robustness to layout diversity by aggregating diverse templates from a fixed era, they fundamentally fail to capture the structural evolution (e.g., complex DOM trees produced by JavaScript frameworks) that emerges over time. We empirically validate this limitation in Appendix F, demonstrating that performance on static snapshots fails to generalize to the structurally evolved live web.
>
> [6] HTTP Archive. The Web Almanac 2024: JavaScript and Markup. HTTP Archive, 2024.

---

### Official Review · Reviewer_uvkz · 2025-10-30

**Soundness:** 3
**Presentation:** 3
**Contribution:** 2
**Rating:** 6
**Confidence:** 3

**Summary:**

This paper introduces LIVEWEB-IE, an online Web Information Extraction (WIE) benchmark, and proposes a multi-stage agentic framework called VGS (Visual Grounding Scraper). By mimicking human cognitive processes, VGS accurately identifies the information that needs to be extracted. Experiments demonstrate that VGS outperforms existing methods on both LIVEWEB-IE and several established WIE benchmarks.

**Strengths:**

The proposed Visual Grounding Scraper (VGS) framework that mimics human information-seeking behavior on web pages is novel and practical. Multi-stage visual grounding (region → element → XPath) effectively reduces HTML noise, achieving great performance on both LIVEWEB-IE and other offline benchmarks.

**Weaknesses:**

1. Weak Motivation. While the paper argues that performance on offline benchmarks fails to generalize to live websites due to temporal changes in web structures, this claim lacks sufficient empirical evidence. For instance, there is no direct comparison showing how existing methods degrade over time on the same websites, nor quantitative data on the frequency or impact of such changes. This undermines the core motivation, as it's unclear whether the offline-to-online gap is as significant as asserted, potentially overstating the need for LIVEWEB-IE.
2. Reproducibility Undermined. Live evaluation causes inconsistent results across runs/time windows, breaking fair comparison and replicability. Website states can vary at each evaluation (e.g., due to updates in layout or content), leading to inconsistent results across runs. Different systems or papers might be tested in non-overlapping time windows, making it impossible to ensure fair comparisons.
3. Expenditure and efficiency in VGS. The multi-stage VGS framework relies heavily on VLMs for visual grounding and pinpointing, which could incur significant computational costs and latency, especially for large-scale scraping. The paper does not discuss efficiency metrics (e.g., inference time or resource usage) or optimizations, making it less practical for real-world deployment compared to simpler HTML-based methods.

**Questions:**

No specific questions. Please refer to the weakness section for detailed concerns.

---

> ### Author Response · Authors · 2025-11-20
>
> *We are grateful for the reviewer's thoughtful comments and helpful feedback. We hope our responses and the accompanying revisions fully address the reviewer's concerns.*
> ### **uvkz-W1**
> >Motivation for LIVEWEB-IE is weak due to insufficient empirical evidence on performance degradation over time.
>
> We thank the reviewer for this constructive criticism. **We conducted a new experiment, which we added to Appendix F, to address this point.**
>
> In this experiment, we utilized the SWDE dataset [1], which consists of HTML snapshots from the early 2010s. From this dataset, we manually curated a new test set by comparing these static snapshots to their corresponding 2025 websites. We specifically selected only those samples where the target value remains present and identical to the one in the original snapshot. We then evaluated baselines and VGS on both the original SWDE snapshots and these corresponding current URLs.
>
> Experimental results provide empirical evidence that performance on an offline snapshot fails to generalize to the current version of the same website. On the GPT-4o model, the F1 scores for HTML-based baselines dropped by an average of 15.14% and VGS dropped by 8.15%. Similarly, on the Qwen2.5-72B model, the baselines saw an average F1 degradation of 17.46%, whereas VGS degraded by 8.63%. This validates our core motivation and demonstrates the need for a benchmark like LiveWeb-IE, which evaluates systems directly on live web pages. Furthermore, it suggests our vision-grounded approach is more robust to the structural evolution.
>
> For a concrete visual illustration of this structural evolution, **we kindly ask the reviewer to also refer to the newly added Fig. 30 and Fig. 31.**
>
> |  |  | | SWDE | | | SWDE-2025 |  |  |
> | :--- | :--- | :--- | :--- | :--- | :--- | :--- | :--- | :--- |
> | **Model** | **Method** | **P** | **R** | **F1** | **P** | **R** | **F1** | **ΔF1** |
> | GPT-4o | CoT | 95.05 | 85.0 | 82.62 | 97.94 | 70.32 | 68.95 | -13.67 |
> | GPT-4o | Reflexion | 91.98 | 95.53 | 90.70 | 98.47 | 73.24 | 73.86 | -16.93 |
> | GPT-4o | AutoScraper | **99.47** | 90.79 | 92.66 | **99.06** | 77.48 | 77.83 | -14.83 |
> | GPT-4o | VGS (ours) | 98.12 | **96.27** | **94.53** | 98.21 | **78.94** | **86.38** | -8.15 |
> | Qwen2.5-72B | CoT | **99.47** | 71.58 | 73.32 | **95.08** | 57.38 | 56.33 | -16.99 |
> | Qwen2.5-72B | Reflexion | 94.62 | 79.74 | 76.67 | 94.12 | 60.89 | 58.79 | -17.88 |
> | Qwen2.5-72B | AutoScraper | 95.57 | 82.11 | 80.0 | 92.40 | 66.16 | 62.58 | -17.51 |
> | Qwen2.5-72B | VGS (ours) | 93.53 | **85.59** | **83.48** | 94.05 | **76.12** | **74.85** | -8.63 |
>
> **Table 1: Empirical validation of the performance gap between static snapshots (SWDE) and live web pages (SWDE-2025). "SWDE-2025" refers to a newly curated test set using the November 2025 web pages corresponding to the original SWDE samples. "$\Delta$F1" denotes the F1 score degradation observed on the live web page compared to the static snapshot.**
>
> [1] Hao et al. From one tree to a forest: a unified solution for structured web data extraction, SIGIR 2011.

---

> ### Author Response · Authors · 2025-11-20
>
> ### **uvkz-W2**
> >Live evaluation undermines reproducibility and fair comparison due to inconsistent results across time windows.
>
> We thank the reviewer for this insightful point, which addresses a fundamental challenge of our new evaluation paradigm.
>
> We respectfully argue that the performance variation the reviewer describes is not a "failure of reproducibility" but rather an **intended and necessary feature** of our benchmark's design. The core motivation for LiveWeb-IE is that a WIE system's performance must be accurately evaluated at the moment of measurement by having it operate in an environment where layouts change over time (**§ 1**).
>
> As the reviewer notes, a system's performance may change when evaluated in a different time window. This is precisely the scenario we aim to measure. If a website's layout evolves and a WIE system's performance degrades, this is not an "inconsistent" or "unreproducible" result. Rather, it is an accurate measurement of that system's current performance and its failure to generalize to the live web's updated structure.
>
> However, to ensure that this live evaluation protocol remains a valid comparison, our benchmark's design is anchored by a critical constraint: the ground-truth content itself is stable (**§ 3.2**). We curate queries for fact-based information that is "unlikely to change over time" (e.g., the 2022 World Cup final results). Therefore, while two systems evaluated in non-overlapping time windows might face different layouts, they are both being tested on the same valid, grounded task of finding the exact same stable ground-truth information.
>
> To clarify our evaluation protocol and its design, **we added a more detailed explanation to § 1 and § 3.**

---

> ### Author Response · Authors · 2025-11-20
>
> ### **uvkz-W3**
> >Discussion on computational costs and efficiency metrics for the VGS framework is missing.
>
> Computational cost, latency, and efficiency are critical points and a key motivation for our adoption of a wrapper generation methodology. The wrapper generation approach is efficient and practical for large-scale scraping because the wrapper is generated once and can be reused across many web pages with similar structures (**§ 1**). We demonstrated this methodological efficiency in **Appendix E**, where VGS is more efficient than the LLM Extractor baseline that incurs substantial inference time for every page.
>
> Regarding the specific point about the wrapper generation cost itself, we conducted a comparative cost analysis against LLM-based wrapper generation baselines. In this experiment, we measured the F1 score and inference time for all methods on our LiveWeb-IE benchmark, segmenting the web pages based on the character length of their HTML as a proxy for complexity. **We added the results to Appendix G.**
>
>
> As shown in the table below, VGS incurs a higher initial time cost than baselines on simple web pages (e.g., <20k chars) due to its visual processing. However, as page complexity increases, the inference time of iterative baselines (Reflexion, AutoScraper) also increases significantly, becoming comparable to VGS. Notably, VGS achieves comparable inference time on complex pages while demonstrating significantly superior extraction accuracy. Given that extracting information from complex and challenging sites is the critical goal, we believe the VGS approach is practical for real-world deployment.
>
> | | Character Length | 0-20k |  | 20k-40k | | 40k-60k |  | 60k-80k |  | 80k+ |  |
> | :--- | :--- | :---: | :---: | :---: | :---: | :---: | :---: | :---: | :---: | :---: | :---: |
> | **Model** | **Method** | **F1 (%)** | **Time (s)** | **F1 (%)** | **Time (s)** | **F1 (%)** | **Time (s)** | **F1 (%)** | **Time (s)** | **F1 (%)** | **Time (s)** |
> | GPT-4o | CoT | 13.22 | 15.16 |34.99 | 10.88 | 31.68 | 16.14 | 19.91 | 17.40 | 6.73 | 20.19 |
> | GPT-4o | Reflexion | 33.61 | 23.75 | 33.70 | 35.62 | 28.64 | 54.90 | 22.97 | 60.17 | 4.87 | 69.28 |
> | GPT-4o | AutoScraper | 35.63 | 25.27 | 29.22 | 44.37 | 33.84 | 51.34 | 20.53 | 59.85 | 7.97 | 75.67 |
> | GPT-4o | VGS (ours) | 64.02 | 42.25 | 47.07 | 54.68 | 52.49 | 62.02 | 42.25 | 64.14 | 35.58 | 72.72 |
> | Qwen-2.5-72B | CoT | 31.21 | 18.54 | 26.19 | 22.73 | 27.44 | 21.28 | 21.79 | 24.43 | 4.66 | 27.44 |
> | Qwen-2.5-72B | Reflexion | 29.47 | 31.87 | 21.99 | 50.35 | 24.31 | 66.48 | 16.90 | 83.95 | 4.58 | 95.71 |
> | Qwen-2.5-72B | AutoScraper | 30.44 | 36.97 | 25.61 | 54.84 | 28.69 | 75.70 | 22.81 | 90.66 | 6.69 | 101.65 |
> | Qwen-2.5-72B | VGS (ours) | 52.37 | 51.89 | 40.34 | 62.44 | 41.89 | 80.36 | 28.40 | 91.17 | 22.59 | 98.83 |
>
> **Table 2. Performance and cost comparison across varying web page complexities.**

---

> ### Author Response · Authors · 2025-11-26
> **A gentle reminder to reviewer uvkz**
>
> Dear reviewer uvkz,
>
> Thank you again for your valuable feedback. As we enter the latter half of the discussion phase, we wanted to ensure our response fully addressed your concerns. In particular, we hope the new experiments on the SWDE dataset (Appendix F) and the cost analysis (Appendix G) effectively resolve your questions regarding the motivation and efficiency of our method. We would be happy to engage in further discussion if you have additional questions.
>
> Best regards,
> Paper 11748 Authors

---

### Official Review · Reviewer_RXSR · 2025-10-31

**Soundness:** 3
**Presentation:** 3
**Contribution:** 3
**Rating:** 6
**Confidence:** 4

**Summary:**

The paper introduces a new benchmark, LiveWeb IE, which focuses on addressing the limitations of previous offline benchmarks that capture only a fixed snapshot in time. LiveWeb IE is claimed to evaluate models directly on live websites, thereby reflecting their performance on temporally evolving web content. The paper further proposes a Web IE framework, Visual Grounding Scraper, which leverages visual cues as guidance rather than directly locating the target element.

**Strengths:**

-Empirical results show that the newly proposed benchmark is more challenging, and the performance on state-of-the-art LLM/LMMs are less saturated; showing a gap between WIE systems and humans on more up-to-date live websites.
-The proposed VGS framework is effective on both closed-source and open-source models
-The paper writing is clear and contains sufficient ablations on the VGS components

**Weaknesses:**

-While the LiveWeb-IE benchmark is claimed to be “evaluating directly against live websites”, it is not clear how the benchmark automatically evolves as the website updates over time. The dataset construction pipeline is still based on a snapshot of a certain time and requires human verification to curate the data. It is potentially an overclaim that the benchmark is “Live”.
-It is also not clear how to handle layout changes through time while still keeping the evaluation/annotation valid; and how much human efforts are required to keep the benchmark up-to-date
-The WIE task is very related to GUI and Computer Use tasks (especially in the settings where the html/a11y tree is available for the perception step); there is a lack of discussion and comparison with the widely studied GUI agent models and literature.

**Questions:**

What is the key difference between the WIE task and a subset of GUI tasks (where the agent do not need to perform action but simply perform the perception)?

---

> ### Author Response · Authors · 2025-11-20
>
> *We sincerely thank the reviewer for valuable comments and for raising important questions. We hope our responses and the accompanying revisions fully address the reviewer's concerns.*
> ### **RXSR-W1&W2**
> >Claim that benchmark is "Live" is potentially an overclaim; details on automatic evolution, layout changes, and human effort are unclear.
>
> We thank the reviewer for raising this important point about the sustainability and definition of our benchmark. We wish to clarify that our benchmark is designed to measure robustness against layout evolution (structural liveness) rather than content updates (informational liveness), ensuring both validity and low maintenance costs.
>
> 1. The "Live" Aspect is the Evaluation Protocol
> Conventional benchmarks evaluate WIE systems on static HTML snapshots captured at a single point in time. The HTML file used for evaluation never changes. In contrast, LiveWeb-IE's evaluation is performed directly against live URLs (**§ 3.1**). When a WIE system is evaluated, it must access the target URL in real-time and process the current DOM structure. This means if a website's layout or structure has been updated since our verification, the system must handle that updated structure (**Fig. 1 - (A)**). This is the fundamental, "Live" difference from static snapshot benchmarks and precisely the real-world challenge we aim to evaluate.
>
> 2. Layout Change is Intended
> The reviewer's question about "how to handle layout changes" is not a flaw, but rather the central purpose of our benchmark's design. As we argued in our Introduction (**§ 1**), the performance of WIE systems is strongly dependent on the structural properties of web pages. Existing static benchmarks, built from HTML snapshots captured at a single point in time (often years in the past), fail to represent the current web page layouts. This leads to the performance generalization gap (**Appendix F, Tab.4**). By forcing WIE systems to evaluate on live URLs, our benchmark evaluates the performance of WIE systems at the specific moment of evaluation.
>
> 3. Evaluation/Annotation Validity is Ensured by Factual Value Stability
> To ensure the long-term validity of our evaluation while capturing this structural dynamism, we separate layout changes from value changes. During data annotation, we explicitly curate queries for attributes whose values are highly unlikely to change over time (**§ 3.2**). We select fact-based, stable information (e.g., the results of the 2022 World Cup final). Therefore, even as a website's layout (e.g., CSS) evolves, the ground-truth value remains factually correct [1]. Our annotation remains valid without requiring continuous human effort to keep the benchmark up-to-date.
>
> In summary, LiveWeb-IE accurately measures the performance of WIE systems on the live web as it exists at the moment of evaluation by requiring systems to access the target URLs. Because our target values are stable, the benchmark does not require continuous human effort to keep the annotations up-to-date.
>
> To clarify this, **we added a more detailed explanation of our benchmark's purpose and design to § 1 and § 3. We also included examples in Fig. 30 and Fig. 31** that illustrate cases where the layout changes while the ground-truth values remain constant.
>
> [1] Mialon et al. GAIA: A Benchmark for General AI Assistants, arXiv'23.11.

---

> ### Author Response · Authors · 2025-11-20
>
> ### **RXSR-W3&Q1**
> >The paper lacks discussion and comparison between the WIE task and the related GUI/Computer Use agent literature, especially regarding the difference from perception-only GUI tasks.
>
> We agree that the tasks are related, as both operate in web environments. However, we believe these two tasks are different in their core objective and approaches.
>
> 1. Difference in Objective
> The focus of GUI agents (operated on the web environment) is on task completion across multiple web pages (e.g., booking a flight) and web element interaction (e.g., click, type) [2,3]. Even within perception-only subtasks, the role of perception is typically intermediate—serving as a prerequisite to determine the next action (e.g., locating a 'login' button to click it) or to assess the state. In contrast, the core objective of WIE is structured data extraction. Here, perception is not a means to an interaction but the end goal itself: to systematically parse a web page and transform its content into a structured format (e.g., extracting a "list of authors").
>
> 2. Difference in Approach
> Different objectives lead to different methodological approaches. GUI agents focus on 'interaction' with the environment and are designed to accurately execute a series of actions like click and type [4,5]. In contrast, WIE systems are designed to accurately identify the specific information requested by the user from within DOM structures and visual layouts, and extract it.
>
> **We supplemented the Related Work (§ 2)** with a comparison to GUI agent literature to more clearly describe the differences in objectives and approaches between the two fields.
>
>
> [2] Deng et al. Mind2web: Towards a generalist agent for the web, NeurIPS 2023 Spotlight.
> [3] Zhou et al. WebArena: A Realistic Web Environment for Building Autonomous Agents, ICLR 2024.
> [4] He et al. WebVoyager: Building an End-to-End Web Agent with Large Multimodal Models, ACL 2024.
> [5] Chae et al. Web-Shepherd: Advancing PRMs for Reinforcing Web Agents, NeurIPS 2025 Spotlight.

---

> > ### Comment · Reviewer_RXSR · 2025-11-26
> >
> > Thanks for the response. If you can address these comments as you wrote in the response, the paper will be clearer and stronger. I will keep my positive score.

---

> > > ### Author Response · Authors · 2025-11-27
> > > **Follow-up response to reviewer RXSR**
> > >
> > > Thank you for your positive feedback and for confirming that our response addressed your concerns.
> > > **We have updated the manuscript to reflect your feedback. We kindly invite you to review the revised version (highlighted in blue), where you can see these changes in context.**
> > >
> > > Specifically, we have:
> > > - Clarified the benchmark's design: Added detailed explanations in **§ 1** and **§ 3** regarding the benchmark's evolution and layout changes.
> > > - Added illustrative examples: Included **Fig. 30** and **Fig. 31** in the Appendix to visually demonstrate stable values amidst layout changes.
> > > - Provided empirical evidence: Added the performance generalization gap analysis in **Appendix F** and **Tab. 4**.
> > > - Expanded related work: Added a comparison with GUI agent literature in **§ 2** to distinguish the WIE task objectives.
> > >
> > > We hope you find that the revised paper successfully reflects your valuable feedback.
> > > Thank you again for your time and insightful comments.

---

> ### Author Response · Authors · 2025-11-26
> **A gentle reminder to reviewer RXSR**
>
> Dear reviewer RXSR,
>
> We appreciate your constructive feedback. In follow-up to our response, we would like to discuss any further questions you might have. We hope that the detailed explanations regarding the live nature of the benchmark (§ 1, 3) and the GUI agent comparison (§ 2) clarify the validity and positioning of our work. Thank you again for your help in strengthening the paper.
>
> Best regards,
> Paper 11748 Authors

---

### Official Review · Reviewer_v3S5 · 2025-11-03

**Soundness:** 3
**Presentation:** 3
**Contribution:** 2
**Rating:** 6
**Confidence:** 4

**Summary:**

The paper tackles the task of Web Information Extraction (WIE) directly on live websites, while prior work has mainly considered static websites.  For this, they introduce a new benchmark LiveWeb-IE which existing Web IE approaches seem to struggle on. The paper also introduces Visual Grounding Scraper (VGS) that leverages VLMs to narrow down the relevant web element to extract desired information. Experiments show that VGS outperforms current WebIE approaches on LiveWeb-IE while also demonstrating improvements on existing WebIE benchmarks.

**Strengths:**

1. The methodology is described well and the paper is easy to read.
2. The experimental results are pretty comprehensive, with a variety of backbone LLMs used.
3. The LiveWeb-IE benchmark can be a very valuable resource to the research community.

**Weaknesses:**

1. The novelty of the VGS approach is limited. The method mainly incorporates VLMs for prompting to narrow down relevant web elements, with the XPath generation part already being done in prior work [1].
2. The related work section is pretty lacking, with no discussion of distinctions/comparisons of VGS with prior WebIE methodologies.
3. While VGS is relatively performant, the authors should also show a cost comparison with prior baselines. The approach of iteratively pass all regions of the webpage to VLM can be cost intensive. The paper does not have any discussion of the additional cost of the proposed approach.

[1] Automating xpath query generation using nlp for streamlined web crawling and gui testing; Kaur et al 2025

**Questions:**

1. Given that the use of live websites within this benchmark creates the problem of information on these websites changing over time, how do the authors  plan to address this? Will the answers regularly be refreshed, similar to FreshQA [2]?

[2] FreshLLMs: Refreshing Large Language Models with Search Engine Augmentation; Vu et al 2023

---

> ### Author Response · Authors · 2025-11-20
>
> *We are grateful for the reviewer's insightful comments and constructive feedback. We hope our responses and the accompanying revisions fully address the reviewer's concerns.*
>
> ### **v3S5-W1**
> >Novelty of the proposed method is limited.
>
> Thank you for your comment on the novelty of our methodology.
>
> We would like to clarify that the most critical challenge in generating robust, accurate XPaths lies in the 'identification' process: precisely locating target elements within web pages while filtering out irrelevant noise. Our fundamental novelty lies in proposing a new vision-based paradigm to address this foundational identification problem, a significant departure from prior HTML-based approaches.
>
> The prior works [1] and [2] also aimed to solve this identification problem, but their novelty lay in HTML-based pre-processing to identify precise information for XPath generation. As we discussed in our Introduction (**§ 1**) and Related Work (**§ 2**) sections, HTML-based approaches struggle with the verbose and noisy nature of HTML.
>
> Instead of relying solely on raw HTML, VGS leverages *'visual information'* to sequentially narrow the relevant region to generate an accurate XPath. This vision-based filtering methodology demonstrates strong performance, outperforming SOTA baselines on both the LiveWeb-IE benchmark (**§ 5.2**) and existing WIE benchmarks (**§ 6.1**). Furthermore, it shows excellent performance on tasks requiring the extraction of images and hyperlinks, not just text (**§ 6.2**).
>
> We believe this **methodological shift** to a vision-based grounding approach, which mimics the human cognitive process of visually identifying information, is the key to its robust performance on these tasks and represents a significant contribution.
>
> ### **v3S5-W2**
> >Related work section is lacking in detail.
>
> We appreciate the constructive feedback regarding the completeness of the Related Work section. **We added a more explicit discussion of methodological distinctions and comparisons in the Related Work section (§ 2).**
>
> We would like to clarify that the core differentiations and comparative analysis between VGS and existing methodologies were described in other sections of the paper:
>
> In the Introduction (**§ 1**), we specified the limitations of existing methods: (1) wrapper-based methods are brittle and require substantial human effort to maintain due to their reliance on structural HTML patterns, and (2) using LLMs directly to extract information from each web page is impractical due to high costs.
>
> Furthermore, we explained the clear distinction from methodologies that generate reusable wrappers using LLMs (**§ 1, 5.1**). Even recent LLM-wrapper methods still use only raw HTML as input [1,2]. In contrast, VGS performs Visual Grounding (**§ 4.2**) and Element Pinpointing (**§ 4.3**) via rendered screenshots. This allows VGS to use a highly refined local HTML segment as input for the final XPath generation step. This *vision-grounded multi-stage filtering approach* is the core differentiator that distinguishes VGS from existing HTML-based methodologies.
>
>
> [1] Kaur et al. Automating xpath query generation using nlp for streamlined web crawling and gui testing, ICTEST 2025.
> [2] Huang et al. AutoScraper: A progressive understanding web agent for web scraper generation, EMNLP 2024.

---

> ### Author Response · Authors · 2025-11-20
>
> ### **v3S5-W3**
>
> >Cost comparison and analysis of the proposed approach is required.
>
> Cost is a critical point and our key motivation for adopting a wrapper generation methodology. The wrapper generation approach is efficient for large-scale tasks because the wrapper can be reused across many web pages with similar structures (**§ 1**). We demonstrated this methodological efficiency in **Appendix E**, where VGS is far more efficient than the LLM Extractor baseline that incurs substantial inference time for every page.
>
> Regarding the reviewer's specific point about the wrapper generation cost itself, **we conducted a comparative cost analysis against LLM-based wrapper generation baselines and added the results to Appendix G.** In this experiment, we measured the F1 score and inference time for all methods on our LiveWeb-IE benchmark, segmenting the web pages based on the character length of their HTML as a proxy for complexity.
>
> As shown in the table below, VGS incurs a higher initial time cost than baselines on simple web pages (e.g., <20k chars) due to its visual processing. However, as page complexity increases, the inference time of iterative baselines (Reflexion, AutoScraper) also increases significantly, becoming comparable to VGS. Notably, VGS achieves comparable inference time on complex pages while demonstrating significantly superior extraction accuracy. Given that extracting information from complex and challenging sites is the critical goal, we believe the VGS approach is practical and scalable.
>
> | | Character Length | 0-20k |  | 20k-40k | | 40k-60k |  | 60k-80k |  | 80k+ |  |
> | :--- | :--- | :---: | :---: | :---: | :---: | :---: | :---: | :---: | :---: | :---: | :---: |
> | **Model** | **Method** | **F1 (%)** | **Time (s)** | **F1 (%)** | **Time (s)** | **F1 (%)** | **Time (s)** | **F1 (%)** | **Time (s)** | **F1 (%)** | **Time (s)** |
> | GPT-4o | CoT | 13.22 | 15.16 |34.99 | 10.88 | 31.68 | 16.14 | 19.91 | 17.40 | 6.73 | 20.19 |
> | GPT-4o | Reflexion | 33.61 | 23.75 | 33.70 | 35.62 | 28.64 | 54.90 | 22.97 | 60.17 | 4.87 | 69.28 |
> | GPT-4o | AutoScraper | 35.63 | 25.27 | 29.22 | 44.37 | 33.84 | 51.34 | 20.53 | 59.85 | 7.97 | 75.67 |
> | GPT-4o | VGS (ours) | 64.02 | 42.25 | 47.07 | 54.68 | 52.49 | 62.02 | 42.25 | 64.14 | 35.58 | 72.72 |
> | Qwen-2.5-72B | CoT | 31.21 | 18.54 | 26.19 | 22.73 | 27.44 | 21.28 | 21.79 | 24.43 | 4.66 | 27.44 |
> | Qwen-2.5-72B | Reflexion | 29.47 | 31.87 | 21.99 | 50.35 | 24.31 | 66.48 | 16.90 | 83.95 | 4.58 | 95.71 |
> | Qwen-2.5-72B | AutoScraper | 30.44 | 36.97 | 25.61 | 54.84 | 28.69 | 75.70 | 22.81 | 90.66 | 6.69 | 101.65 |
> | Qwen-2.5-72B | VGS (ours) | 52.37 | 51.89 | 40.34 | 62.44 | 41.89 | 80.36 | 28.40 | 91.17 | 22.59 | 98.83 |
>
> **Table 1. Performance and cost comparison across varying web page complexities.**

---

> ### Author Response · Authors · 2025-11-20
>
> ### **v3S5-Q1**
> >Given that the use of live websites within this benchmark creates the problem of information on these websites changing over time, how do the authors plan to address this?
>
> This is a critical point that we considered deeply, as ensuring evaluation validity is paramount. The core of this issue lies in distinguishing between two types of temporal change: (1) changes to the **factual value** of an attribute, and (2) changes to the **layout and structure** of the web page.
>
> As the reviewer rightly notes, if the ground-truth value for a query on a web page changes, our annotation would become incorrect. This is the challenge addressed by datasets like FreshQA [3], which focus on changing values.
>
> However, the purpose of LiveWeb-IE is different. Given that WIE systems are sensitive to web page structure, our benchmark is designed to evaluate performance directly against the current structure of a live web page using its URL (**§ 1, 3**). To ensure the evaluation remains valid while capturing this structural dynamism, we only select attributes where the value itself is temporally stable (**§ 3.2**). For instance, a query about the results of the '2022 World Cup final' targets a static fact. While the website's layout (CSS, DOM structure) may evolve over time, the value (Argentina) itself will not change.
>
> This design allows us to stably evaluate a WIE system's information extraction performance on web pages as they exist at the current moment. Because our ground-truth values are curated for stability, we do not require the continuous refreshment protocol.
>
> To help understand that our benchmark data remains valid over time, **we included examples in Fig. 30 and Fig. 31** that illustrate cases where the layout changes while the ground-truth values remain constant.
>
> [3] Vu et al. FreshLLMs: Refreshing Large Language Models with Search Engine Augmentation, ACL 2024 Findings.

---

> ### Author Response · Authors · 2025-11-26
> **A gentle reminder to reviewer v3S5**
>
> Dear reviewer v3S5,
>
> We are grateful for your valuable review. Following our response, we would like to check if our rebuttal has sufficiently addressed your concerns. We hope that the expanded discussion on methodological differences (§ 2), the efficiency analysis (Appendix G), and the clarification regarding the temporal stability of ground-truth values (§ 1, 3) have resolved your questions. Please let us know if there are any remaining concerns we can address.
>
> Best regards,
> Paper 11748 Authors

---

### Author Response · Authors · 2025-11-20
**General Responses to all reviewers**

*We sincerely thank all four reviewers for careful reading, constructive feedback, and valuable suggestions.*

In response, we revised the manuscript to address the points raised and enhance its clarity, highlighting all changes in blue. We also provide detailed responses to each point below. We hope the reviewers find the revisions and our accompanying responses helpful.

**We are pleased that the reviewers recognized the key strengths of our work:**

- **Clarity and Comprehensive Evaluation:** Reviewers commended the paper's clarity, readability, comprehensive experimental results, and sufficient ablation studies. [v3S5, RXSR]

- **Contribution of Benchmark:** Reviewers highlighted that the benchmark is a valuable, challenging, and less-saturated resource, noting its solid engineering pipeline and systematic four-type task definition. [v3S5, RXSR, D9Lx]

- **Novelty of Methodology:** Reviewers recognized the VGS framework as novel and practical, noting its multi-stage visual grounding effectively reduces noise and demonstrates strong performance across diverse models. Its role as a baseline was also valued for illustrating the benchmark's intended use and providing an initial reference for comparison. [RXSR, uvkz, D9Lx]


**Our responses to the key concerns, along with the corresponding manuscript revisions, are as follows:**

- **Novelty and Positioning of Methodology:** We clarified that the core novelty of VGS is not the final XPath synthesis, but the vision-based identification paradigm used to find the target elements. **We updated the Related Work (§ 2)** to explicitly differentiate this vision-based approach from prior HTML-based methods and to clarify the distinct goals of WIE versus GUI agents. [v3S5-W1&W2, RXSR-W3&Q1]

- **Cost and Efficiency of Methodology:** To address concerns regarding computational cost and efficiency, **we performed a efficiency analysis (Appendix G, Fig.10).** While VGS is a wrapper generator, making it highly efficient for large-scale scraping, we analyzed the generation cost itself (Appendix E). The results show that for complex web pages, VGS is more efficient than HTML-only LLM baselines, as its visual filtering scales better than parsing token-heavy HTML. [v3S5-W3, uvkz-W3]

- **Motivation for the Benchmark:** To provide empirical evidence for our benchmark motivation, **we conducted a new experiment (Appendix F, Tab.4).** We evaluated methods on original SWDE snapshots from the early 2010s and then on their corresponding live 2025 URLs. The results show a performance degradation on the live URLs, which validates our claim that performance on outdated snapshots fails to generalize. [uvkz-W1]

- **Benchmark Design - "Live" Evaluation and Stability:** We clarified a critical distinction in our benchmark's design: we decoupled the stability of values from the stability of web page structures. LiveWeb-IE forces systems to process the current web structure—allowing for an accurate evaluation of their performance on the web page as it exists at that moment—while the stable values ensure the evaluation task itself remains valid. **We updated the Introduction (§ 1) and Dataset (§ 3), and added figures (Fig. 30, 31)** to make this critical point explicit. [v3S5-Q1, RXSR-W1&W2, uvkz-W2, D9Lx-W1&W2&Q1&Q2]

- **Motivation for Multimodality:** We clarified that multimodality (text, images, hyperlinks) is critical for a benchmark aimed at real-world scenarios, where user queries are not limited to text. Our analysis (§ 6.2) confirms that non-textual extraction remains a key challenge where models struggle, justifying its inclusion as a core evaluation component. **We added clarifications on this point to the Introduction (§ 1), Dataset (§ 3.2), and Discussion (§ 6.2) sections.** [D9Lx-W3&Q3]

If there are remaining questions or any additional comments, we are happy to address them within the remaining discussion time.
Thank you again for your thoughtful reviews.

Best regards,
Paper 11748 Authors

---

### Author Response · Authors · 2025-12-03
**Summary of the Revised Sections**

*We thank the reviewers for their valuable feedback, which helped us improve the quality of our paper.*

We outline the key updates in the revised manuscript below to assist in locating the revisions corresponding to our responses.

**§1. Introduction**
- Substantiated the necessity of live evaluation by citing empirical evidence (uvkz-W1) and clarified the benchmark design principles (v3S5-Q1, RXSR-W1&W2, uvkz-W2, D9Lx-W1&Q1&Q2).
- Highlighted the importance of multimodality (D9Lx-W3&Q3).

**§2. Related Work**
- Distinguished WIE from general web agent tasks (RXSR-W3&Q1) and our vision-based approach from HTML methods (v3S5-W2).

**§3. LIveWeb-IE**
- Clarified the distinction between structural liveness and informational stability to justify the benchmark design (v3S5-Q1, RXSR-W1&W2, uvkz-W2, D9Lx-W1&Q1&Q2).

**§3.2. Dataset Construction**
- Detailed the attribute selection rationale, ensuring evaluation stability (v3S5-Q1, RXSR-W1&W2, uvkz-W2, D9Lx-W1&W2&Q1&Q2) and explained the functional integration of non-textual attributes into the ground truth (D9Lx-W3&Q3).

**§6.2. Performance Analysis by Data Category**
- Emphasized the role of multimodality in reflecting the diverse requirements of real-world scraping scenarios (D9Lx-W3&Q3).

**Appendix F. Performance Gap Between Static Snapshots and Live Web Pages**
- Added experiments quantifying performance degradation from static snapshots to live pages (uvkz-W1).
- Tab. 4

**Appendix G. Efficiency Comparison with LLM-based Wrapper Generation Baselines**
- Added cost-performance analysis across varying web page complexities (v3S5-W3, uvkz-W3).
- Fig. 10

**Appendix K. Limitations**
- Discussed the benchmark's scope regarding temporally stable information and dynamic data (D9Lx-W2).

**Appendix Figures**
-  Fig. 30 & 31: Visualized the distinction between structural liveness and informational stability (v3S5-Q1, RXSR-W1&W2, uvkz-W1, D9Lx-W1&Q1).

---

### Meta-Review · Area_Chair_hHby · 2026-01-08

**Summary:**

The paper introduces LiveWeb-IE, a benchmark designed for Web Information Extraction (WIE) that evaluates systems against live URLs rather than static HTML snapshots. The goal is to capture the temporal evolution of web layouts. The authors also propose a Visual Grounding Scraper (VGS) as a baseline, which uses visual cues to narrow down elements before extraction.

**Reviewer Concerns:**

The reviewers raised several valid concerns initially:

- Validity & Reproducibility (v3S5, RXSR, D9Lx): There was confusion about how a "live" benchmark works if the content changes. Authors clarified that they track "stable values" (facts that don't change, e.g., "2022 World Cup winner") even if the page layout (DOM/CSS) evolves. This distinction between structural liveness and informational stability was crucial.


- Motivation (uvkz, D9Lx): Reviewers questioned if live evaluation is actually harder or necessary compared to static snapshots. The authors added a strong experiment (Appendix F) comparing old SWDE snapshots to their 2025 live versions, showing a significant performance drop (~15% drop for GPT-4o). This empirically proves that static benchmarks don't fully generalize to the modern web.

- Cost (v3S5, uvkz): Concerns about the efficiency of the vision-based VGS. Authors added Appendix G, showing that while VGS has a startup cost, it scales better or comparably to DOM-heavy baselines on complex pages.


- Multimodality (D9Lx): Unclear role of images/links. Authors clarified this is essential for real-world queries and showed performance dips on non-text tasks.


**Addressed**: I believe the concerns regarding validity, motivation (empirical evidence), and cost were well-addressed by the new experiments and clarifications. Reviewer RXSR explicitly noted the paper would be "clearer and stronger" after these changes.


**Outstanding**: Reviewer D9Lx remained skeptical about the necessity of live evaluation, arguing a sufficiently diverse static benchmark could achieve the same goal. While the authors argued that static benchmarks can't predict future structural paradigms (like new JS frameworks), this remains a philosophical disagreement on benchmark design.

**Reviewer Scores:**

- Reviewer v3S5 (Current: 6): Score: 6 (but closer to 8). The reviewer asked for cost analysis and clarification on the "changing info" problem. The authors provided the cost table and the "stable value" explanation. I think they would have bumped their score.


- Reviewer RXSR (Current: 6): Score: 6  (but closer to 8). This reviewer explicitly stated: "If you can address these comments... the paper will be clearer and stronger. I will keep my positive score". Given the strong rebuttal, they likely would have championed the paper more strongly.

- Reviewer uvkz (Current: 6): Score: 6  (but closer to 8). Their main issue was "Weak Motivation" and lack of evidence for the offline-to-online gap. The new experiment in Appendix F directly quantified this gap, effectively resolving their primary weakness.


- Reviewer D9Lx (Current: 4): Score: 6. They acknowledged the rebuttal improved the framing and multimodality sections. While they stuck to their guns about static benchmarks being sufficient, they admitted the clarifications were helpful. They likely would have moved to a borderline/weak accept.

---

### Decision · Program_Chairs · 2026-01-26

Accept (Poster)